# Model Fusion through Bayesian Optimization in Language Model Fine-Tuning

**Chaeyun Jang**[*]
KAIST
jcy9911@kaist.ac.kr

**Hyungi Lee**[*]
KAIST
lhk2708@kaist.ac.kr

**Jungtaek Kim**[†]
University of Pittsburgh
jungtaek.kim@pitt.edu

**Juho Lee**[†]
KAIST
juholee@kaist.ac.kr

## Abstract

Fine-tuning pre-trained models for downstream tasks is a widely adopted technique known for its adaptability and reliability across various domains. Despite its conceptual simplicity, fine-tuning entails several troublesome engineering choices, such as selecting hyperparameters and determining checkpoints from an optimization trajectory. To tackle the difficulty of choosing the best model, one effective solution is *model fusion*, which combines multiple models in a parameter space. However, we observe a *large discrepancy between loss and metric landscapes* during the fine-tuning of pre-trained language models. Building on this observation, we introduce a novel model fusion technique that optimizes both the desired metric and loss through *multi-objective Bayesian optimization*. In addition, to effectively select hyperparameters, we establish a two-stage procedure by integrating Bayesian optimization processes into our framework. Experiments across various downstream tasks show considerable performance improvements using our Bayesian optimization-guided method. Code will be available at: https://github.com/chaeyoon-jang/bomf.git.

## 1 Introduction

The field of Natural Language Processing (NLP) has significantly advanced with the pre-training of Transformer-based models on large amounts of texts without supervision. In general, these pre-trained networks are fine-tuned on supervised downstream tasks to solve particular tasks. The rise of Large Language Models (LLMs) such as GPT [50] and LLaMA [63] has increased demands for huge memory and computing during fine-tuning on downstream tasks. In response, low rank approximation methods such as Low-Rank Adaptation (LoRA) [22] and Quantized Low-Rank Adaptation (QLoRA) [11] have been adopted recently to fine-tune the LLM. However, the fine-tuning of Pretrained Language Models (PLMs) exhibits high sensitivity to marginal variations in hyperparameters such as learning rate and batch size, often leading to training failure and the performance drop of a fine-tuned model [45], while searching hyperparameters requires a vast amount of resources.

An effective strategy to seek an optimal model among multiple candidates is model ensembling, which yields impressive performance on generalization and robustness [33]. However, traditional ensemble methods lead to several drawbacks including the space and computational costs that linearly scale with the number of models involved. These issues are particularly pertinent for LLMs, since individual

---

[*]Co-first authors
[†]Co-corresponding authors

38th Conference on Neural Information Processing Systems (NeurIPS 2024).

models are costly to train and test. Therefore, we can alternatively utilize model fusion to aggregate multiple models into a single proficient model on a parameter space. One of its simplest forms, known as Stochastic Weight Averaging (SWA) [25], involves taking the average of model parameters obtained during an optimization process. Despite its simplicity, SWA and its variants have proven successful across various tasks, notably in computer vision [25, 42, 6, 46]. Recent advancement in this field is the concept of Model Soups, which has been introduced by Wortsman et al. [70]. This approach weight-averages a set of models, obtained from multiple fine-tuning runs with different hyperparameters to create a powerful model that outperforms both individual and ensemble models.

The effectiveness of model fusion has predominantly been explored in the visual domain. For instance, while Model Soups have shown considerable improvements in image classification, they have not demonstrated superiority in the NLP tasks [70]. The existing averaging methods like SWA make use of their ability to encourage a fused model to locate on the center of the flatter area near local optima [25, 20], as loss landscapes are analogous to metric landscapes in computer vision tasks. Unfortunately, for PLMs, loss landscapes are substantially mismatched to metric landscapes, so that the flat loss minimum found by SWA does not necessarily correspond to the flat metric minimum making a simple averaging method fail to find a superior model.

In this paper, we present a novel model fusion approach with an efficient hyperparameter selection strategy, denoted as Bayesian Optimization Model Fusion (BOMF), specifically designed to fine-tune PLMs. To motivate our method, we start by illustrating two empirical analyses. Firstly, we demonstrate that the existing model fusion techniques are not suitable for PLMs. Secondly, we highlight that the optimal hyperparameters for PLMs exhibit consistency on varying the number of frozen layers or the rank used in the LoRA setting [22].

Based on these findings, we introduce our proposed method to build a single model, aggregated through the weighted combination of individual models. Supposing that evaluation metrics are non-differentiable, we employ Bayesian Optimization (BO) [5, 18], which is a black-box optimization technique, in developing our model fusion method. To the best of our knowledge, this is the first study that utilizes BO in the context of model fusion, in order to achieve the following objectives:

- **Utilization of Both Metrics and Loss Functions in Model Fusion.** Instead of running BO with an averaged target metric, we use Multi-Objective Bayesian Optimization (MOBO) that considers both metrics and loss functions for model fusion. Despite low correlations between loss and metric values, we find that incorporating loss values still serves as useful guidance.

- **Two-Stage Model Fusion.** We devise our model fusion process as a two-stage BO procedure. One is for optimizing hyperparameters in fine-tuning and the other is dedicated to our model fusion method. The objective of the first stage is to maximize gains from the second stage to find hyperparameters leading to the optimal fused model after the BO of the second stage.

We demonstrate the effectiveness of BOMF on several NLP tasks, including both Natural Language Understanding (NLU) and Natural Language Generation (NLG), with ROBERTa, Text-to-Text Transfer Transformer (T5) and LLaMA. Through these comprehensive experiments, we assess the performance of BOMF in diverse NLP tasks and uncover the interesting properties of our approach through various ablation studies.

## 2    Preliminaries

**Problem Setup.**    In this paper, we explore the process of fine-tuning PLMs using two types of datasets: a downstream training dataset $\mathcal{D}_{\text{trn}}$ and a validation dataset $\mathcal{D}_{\text{val}}$. Assuming that we are given a *pre-trained* set of weights $\boldsymbol{\theta}_{\text{init}}$ and a *trainable* set of weights $\mathbf{w}_{\text{init}}$ for our PLM denoted as $\mathcal{M}(\boldsymbol{\theta}, \mathbf{w})$, $\mathbf{w}_{\text{init}}$ is either a subset of $\boldsymbol{\theta}_{\text{init}}$ or LoRA weights [22]. Specifically, in the former case, $\mathbf{w}_{\text{init}}$ is deliberately selected from $\boldsymbol{\theta}_{\text{init}}$. As a special case, $\mathbf{w}_{\text{init}}$ will be identical to $\boldsymbol{\theta}_{\text{init}}$ if any layers or weights are not frozen. Meanwhile, if the LoRA method is employed during the fine-tuning of our model, $\mathbf{w}_{\text{init}}$ will be the LoRA weights.

We use $K$ distinct metrics, denoted as $f_{\text{metric}}^{(k)}(\mathcal{M}, \mathcal{D})$ for $k \in [K]$, to evaluate our model's performance on a given task. Each metric $f_{\text{metric}}^{(k)}$ is typically non-differentiable, while a differentiable loss function $f_{\text{loss}}$ is employed for training. Assuming that $\mathcal{D}_{\text{val}}$ is similar to the true data distribution, our goal is to

find the optimal set of trainable weights $\mathbf{w}^\star$ that minimizes the following:

$$\mathbf{w}^\star = \underset{\mathbf{w} \in \mathbf{W}}{\arg\min} \sum_{k=1}^{K} \bar{f}_{\text{metric}}^{(k)}(\mathcal{M}(\boldsymbol{\theta}_{\text{init}}, \mathbf{w}), \mathcal{D}_{\text{val}}), \qquad (1)$$

where $\bar{f}_{\text{metric}}^{(k)}$ is a normalized version of $f_{\text{metric}}^{(k)}$ for all $k \in [K]$, and $\mathbf{W}$ is the space of trainable weights. However, due to the non-differentiability of the $f_{\text{metric}}^{(k)}$ functions, conventional approaches resort to finding approximate solutions using gradient descent or its variants, as shown below:

$$\tilde{\mathbf{w}}^\star = \underset{\mathbf{w} \in \mathbf{W}}{\arg\min} \, f_{\text{loss}}(\mathcal{M}(\boldsymbol{\theta}_{\text{init}}, \mathbf{w}), \mathcal{D}_{\text{trn}}). \qquad (2)$$

As will be discussed in the subsequent section, misalignment between loss and metric surfaces is more prominent in PLMs compared to computer vision models. To address this challenge, we propose a novel method to more adequately make use of $\{f_{\text{metric}}^{(k)}\}_{k=1}^{K}$ and $\mathcal{D}_{\text{val}}$ by considering Equation 1.

**Model Fusion.** In the recent work [25, 70, 53, 54], there has been a growing interest in the use of *weight averaging* or *model fusion* across diverse tasks. This line of research is an effective strategy to achieve superior performance in downstream tasks, all while managing computational costs by aggregation of multiple models. In this context, the aggregation of multiple models involves the identification of a set of $N$ fine-tuned trainable weights, denoted as $\mathcal{S} = \{\mathbf{w}_i\}_{i=1}^{N}$. The objective is to derive a fused weight vector, $\bar{\mathbf{w}}$, by utilizing $\mathcal{S}$, such that $\bar{\mathbf{w}}$ outperforms all other members in $\mathcal{S}$. This can be expressed as $L_{\text{metric}}(\bar{\mathbf{w}}) \leq \arg\min_{\mathbf{w} \in \mathcal{S}} L_{\text{metric}}(\mathbf{w})$, where $L_{\text{metric}}(\mathbf{w}) := \sum_{k=1}^{K} \bar{f}_{\text{metric}}^{(k)}(\mathcal{M}(\boldsymbol{\theta}_{\text{init}}, \mathbf{w}), \mathcal{D}_{\text{val}})$.

Model fusion approaches can be categorized into two main types: 1) uniform averaging and 2) weighted averaging. Uniform averaging methods, e.g., SWA [25], Greedy Soups [70], involve the straightforward process of uniformly averaging weights within a subset $\bar{\mathcal{S}} \subseteq \mathcal{S}$ to obtain an improved performing weight vector $\bar{\mathbf{w}}$, i.e. $\bar{\mathbf{w}} = \frac{1}{|\bar{\mathcal{S}}|} \sum_{\mathbf{w} \in \bar{\mathcal{S}}} \mathbf{w}$. Here, selecting a suitable subset $\bar{\mathcal{S}}$ is an important strategy for each method. On the other hand, weighted averaging approaches, e.g., Learned Soups [70] and Rewarded Soups [54], aim to determine an optimized weight vector $\mathbf{w}$ by forming a convex combination of parameters from $\mathcal{S}$, expressed as $\bar{\mathbf{w}} = \sum_{i=1}^{N} \delta_i \mathbf{w}_i$, where each averaging coefficient $\delta_i$ satisfies $\delta_i \geq 0$, and $\sum_{i=1}^{N} \delta_i = 1$. While weighted averaging methods offer more flexibility compared to uniform averaging, they often require additional training to determine suitable values for the coefficient set $\boldsymbol{\delta}$ through gradient descent updates based on the loss function $f_{\text{loss}}$. However, in our proposed method, we suggest a weighted averaging technique that considers not only the loss function $f_{\text{loss}}$ but also the metrics $\{f_{\text{metric}}^{(k)}\}_{k=1}^{K}$.

**Multi-Objective Bayesian Optimization.** BO is a sample-efficient black-box optimization technique with probabilistic regression. Since we assume that an objective to optimize is unknown, a surrogate function, which is generally a probabilistic regression model, is estimated instead. The key desired properties of the surrogate function are attained by considering how a search space is exploited and explored through its outputs. Utilizing the surrogate function, BO eventually optimizes a specific form of optimizable function, called an acquisition function; see [5, 18] for details.

On top of generic BO, MOBO is used to solve an optimization problem, involved with $K$ different competing objectives:

$$\mathbf{x}^\dagger = \underset{\mathbf{x}}{\arg\min}(f_1(\mathbf{x}), f_2(\mathbf{x}), \ldots, f_K(\mathbf{x})). \qquad (3)$$

Supposing that we cannot directly access $f_1, f_2, \ldots, f_K$, probabilistic surrogate models, which are alternatives to unknown objectives, should be used to determine a next point to evaluate. To find a solution candidate of Equation 3 using MOBO, we can consider scalarization of either the realizations of surrogate models or acquisition functions corresponding to multiple objectives [48]. In contrast to the scalarization method, the maximization of Expected HyperVolume Improvement (EHVI), on a metric space [16] can be used:

$$\mathbf{x}^\dagger = \underset{\mathbf{x}}{\arg\max} \, \text{EHVI}(\mathbf{x}; \mathbf{Y}, \mathbf{r}), \qquad (4)$$

where a hypervolume is defined as the size of space between the Pareto frontier of $n$ historical evaluations $\mathbf{Y} \in \mathbb{R}^{n \times K}$ and a reference point $\mathbf{r}$. While the scalarization determines query points

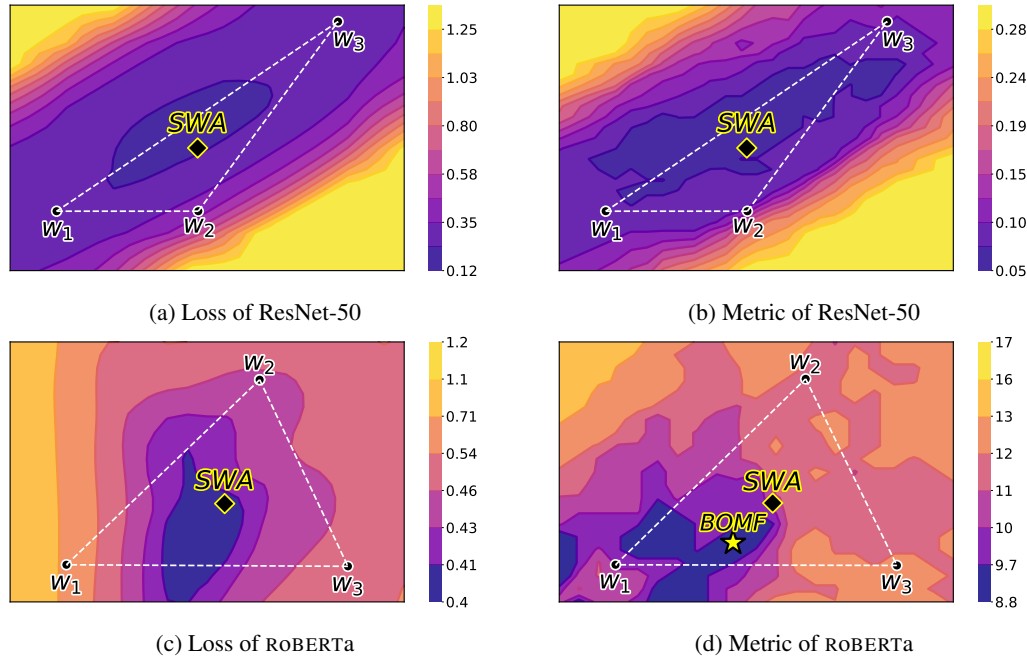

(a) Loss of ResNet-50        (b) Metric of ResNet-50

(c) Loss of ROBERTa        (d) Metric of ROBERTa

Figure 1: Visualization of the loss landscape over parameters (Figures 1a and 1c) and the metric landscape over parameters (Figures 1b and 1d) for both the vision task (Figures 1a and 1b) and the NLP task (Figures 1c and 1d). The metric is $1-$accuracy and F1 score for the vision task and the NLP task, respectively. In the vision task, we fine-tune the ResNet-50 model [21] pre-trained with ImageNet-21k [56] on the Caltech-101 dataset [35], while in the NLP task, fine-tuning was performed on the pre-trained ROBERTa model on the MRPC dataset. The members of the SWA for each figure are denoted as $w_1, w_2, w_3$.

by aggregating $K$ outputs with particular (potentially random) coefficients, the hypervolume improvement maximization chooses query points that widen the expected hypervolume, which is more robust to function scales without the sampling distributions of scalarization coefficients. As reported in the previous work [10, 3], compared to other existing MOBO algorithms, $q$NEHVI which is a variant of the EHVI method that evaluates a batch of $q$ points in a parallel manner. Building on the powerful MOBO algorithm, our model fusion framework is capable of determining averaging coefficients efficiently reducing the number of evaluations required to find better fused PLMs.

## 3 Empirical Findings

In this section, we present empirical observations motivating our model fusion strategy. In § 3.1, we initially illustrate distinct findings: unlike in computer vision tasks, in NLP tasks, there exists a significant misalignment between the loss and metric surfaces. This misalignment poses a challenge for straightforward model fusion methods when fine-tuning PLMs. In § 3.2, we find that the optimal fine-tuned hyperparameters for PLMs analogously align across different architectural configurations varying the number of frozen layers or variations in rank in the LORA setting.

### 3.1 On Misalignment in Loss and Metric Landscapes

The well-known success of uniform averaging, e.g., SWA and Model Soups, in image classification tasks, has been grounded on the flatness of a loss landscape. As one can see in Figure 1a, the use of uniform averaging successfully explores minima on the flatter region of the loss landscape using individual weights close to the flatter region, resulting in enhanced generalization loss on a test dataset. This generalization effect is similarly observed in the case of the metric landscape, as illustrated in Figure 1b, owing to the similarity between the loss and metric landscapes. This similarity is the consequence of the inherent similarity between the loss function and the metric [43]. However, the

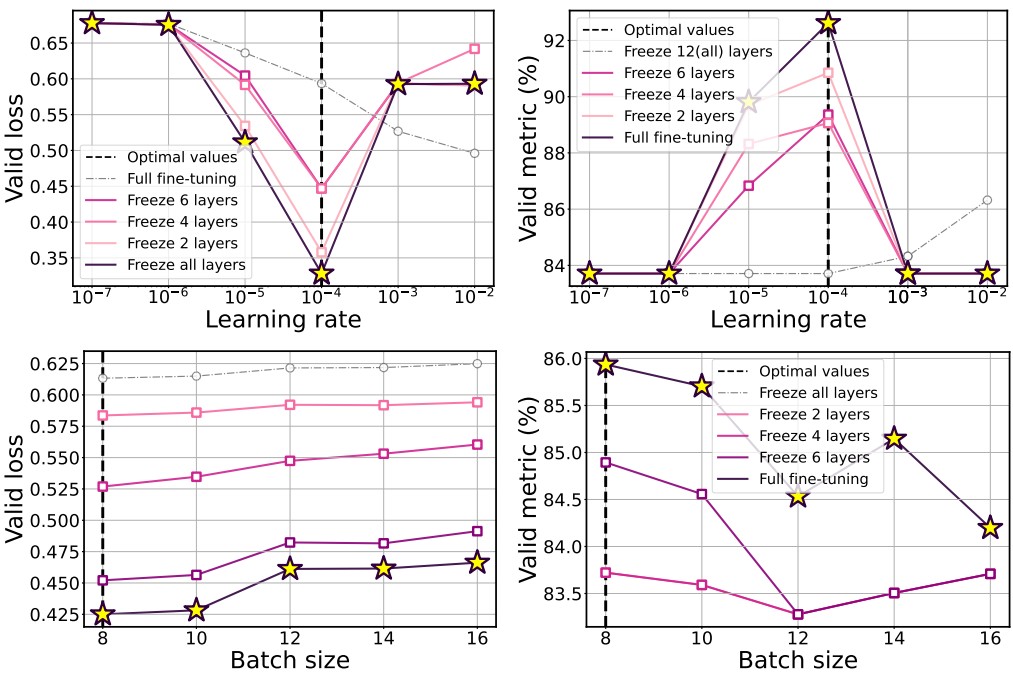

Figure 2: Validation results on the MRPC dataset for RoBERTa: loss (shown in left panels) and F1 score (in right panels) for varying learning rates, batch sizes, and frozen layers. Optimal hyperparameters align well across different frozen layers, except when all pre-trained layers are frozen.

domain of language modeling, characterized by semantic, morphosyntactic, and pragmatic intricacies, requires the evaluation of generalization performance across a diverse array of tasks and metrics [12]. It is unlikely to precisely align these metrics with a training loss function [74, 39], leading to a misalignment that often results in more complex and less flat surfaces in language tasks compared to the loss function visually demonstrated in Figures 1c and 1d.

In Figures 1c and 1d, we find that while uniform averaging can reach high generalization performance based on the loss function, it poorly performs concerning the metric function compared to the best-performing weight in $\mathcal{S}$. However, Figure 1d shows that even though the uniform averaging of three weight points degrades the metric performance, better points in terms of higher metric values exist in the convex set of the three weight points. The empirical results we observe above, which are caused by the complex and misaligned surface, motivate the need to utilize weighted averaging methods and seek the optimal combination of averaging weights based on the metric. This does not agree with the previous findings in vision tasks [70] and Figure 1b which argue minimal performance difference between the weighted averaging and the uniform averaging. Refer to Appendix C.1 for numerical results that show the discrepancy between the loss and metric landscapes in PLMs.

## 3.2 On Hyperparameter Alignment

Discovering the optimal training hyperparameters incurs significant computational costs, particularly when fine-tuning extensive foundational models [2, 45, 66]. This challenge arises since the ideal set of hyperparameters tends to vary in tandem with changes in both tasks and model structures.

Surprisingly, our empirical findings reveal a consistent alignment of optimal hyperparameters when fine-tuning PLMs, regardless of variations in the number of frozen layers or the rank of LoRA. As illustrated in Figure 2, the alterations in validation loss and metric resulting from changes in the learning rate or batch size exhibit a similar pattern across different numbers of frozen layers, except in the case when all pre-trained layers are frozen and only the classifier layer is trained. This proves that we can decrease computational cost for searching the optimal hyperparameters by tuning on smaller models with more frozen layers or LoRA with smaller ranks. Refer to Appendix C.2 to see the additional results when varying the adam beta, learning rate schedule, as well as the case of the LoRA.

Yang et al. [72] demonstrate that employing a particular model weight initialization method and learning rate scheduling method, referred to as $\mu$-parametrization, enables the transferability of certain training hyperparameters (such as learning rate and momentum) varying the width of the model. However, it is important to note that these results specifically pertain to scenarios where models are trained from scratch. This distinction is noteworthy as our context involves the fine-tuning of PLMs. It would be a great future research direction to theoretically analyze this phenomenon.

## 4 Bayesian Optimization Model Fusion

In this section, BOMF unfolds in three key steps. In § 4.1, we present the process of constructing a set of fine-tuned trainable weights $\mathcal{S}$, serving as components for model fusion. In § 4.2, we introduce a method to identify optimal hyperparameters crucial in the construction of the set $\mathcal{S}$ based on the findings explained in § 3.2. Finally, we delve into how we conduct weighted averaging in § 4.3, following the insights discussed in § 3.1.

### 4.1 Fusion Member Sampling

To improve the performance of our model through model fusion, it is crucial to carefully create the set $\mathcal{S}$ by employing an appropriate weight sampling method. There are two main types of weight sampling methods: 1) sampling from multiple training trajectories [70] and 2) sampling from a single training trajectory with proper learning rate scheduling [25]. However, Wortsman et al. [70] indicate that, when applying model fusion with samples from multiple training trajectories, the performance improvement becomes less significant during the fine-tuning of PLMs compared to vision tasks. This limitation in NLP tasks is attributed to the misalignment in loss and metric surfaces, as discussed in § 3.1. Furthermore, when employing multiple training trajectories to sample fusion members, the training computation cost increases linearly in proportion to the number of fusion members. This poses a significant challenge, particularly in the context of fine-tuning PLMs. For these reasons, in our approach, we collect our fusion members from a single training trajectory. Since the fine-tuning process of PLMs involves a small number of training epochs and exhibits rapid convergence [41], we start gathering fusion members after 50% of the training epochs are completed. This timing is slightly quicker than the point described in the work [25], which begins collecting after 75% of the training epochs are concluded. Once we start collecting the fusion members, we proceed to uniformly sample 15 members throughout the remaining training epochs. Refer to Appendix A for more details on the process of collecting fusion members.

### 4.2 Hyperparameter Search via Bayesian Optimization

In the construction of a set of fusion members $\mathcal{S}$ from a single training trajectory, the effectiveness of the training trajectory significantly impacts the ultimate metric performance of the fused model weight $\bar{\mathbf{w}}$. In this context, the effectiveness of a training trajectory refers to the model's metric performance using the best-performing weight within that trajectory on the validation dataset $\mathcal{D}_{\text{val}}$. The correlation in Figure 3 strongly indicates that the performance of the best-performing weight is positively correlated with the performance of the fused weight. Consequently, to achieve the best performance of the fused weight, it becomes crucial to identify the set of optimal hyperparameters $\boldsymbol{\lambda}$ that results in the most effective training trajectory. However, two primary challenges arise when searching for the optimal hyperparameters $\boldsymbol{\lambda}^{\star}$ that yield the best metric

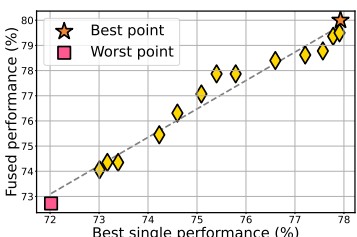

Figure 3: Correlation between the performance of best-performing weights in a training trajectory and the performance of the fused model. We fine-tune the ROBERTa model on the RTE dataset. Each point is obtained from the evaluation of a single trajectory with varying hyperparameters.

performance: 1) the metric functions $\{f_{\text{metric}}^{(k)}\}_{k=1}^{K}$ are non-differentiable and 2) we need to efficiently assign computational resources in finding better hyperparameters beyond naïve methods such as grid search. To remedy these two issues simultaneously, in BOMF, we employ BO to find the optimal set

of hyperparameters:

$$\boldsymbol{\lambda}^* = \arg\min_{\lambda} \sum_{k=1}^{K} \bar{f}_{\text{metric}}^{(k)}(\mathcal{M}(\boldsymbol{\theta}_{\text{init}}, \mathbf{w}(\boldsymbol{\lambda})), \mathcal{D}_{\text{val}}), \tag{5}$$

where $\mathbf{w}(\boldsymbol{\lambda})$ represents the best-performing weight within the training trajectory associated with the hyperparameter set $\boldsymbol{\lambda}$. Here, we utilize Gaussian process (GP) regression [55] and Log-Expected Improvement [3] as a surrogate function and an acquisition function, respectively. We employ three randomly initialized sets of hyperparameters as the starting point for BO, conducting 10 iterations of computations to determine the optimal set $\boldsymbol{\lambda}^\star$.

The sequential nature of BO computations can lead to a substantial computational load, particularly in the context of fine-tuning PLMs. To address this issue and propose a more computationally efficient BO approach, we draw insights from the observations discussed in § 3.2. The alignment of the best hyperparameters for fine-tuning between the full model and lightweight models (e.g., frozen layers model or reduced rank LORA) allows us to utilize the lightweight model instead of the full model when seeking the optimal set $\boldsymbol{\lambda}^\star$ as follows:

$$\boldsymbol{\lambda}^\star = \arg\min_{\lambda} \sum_{k=1}^{K} \bar{f}_{\text{metric}}^{(k)}(\mathcal{M}(\boldsymbol{\theta}_{\text{init}}, \widehat{\mathbf{w}}(\boldsymbol{\lambda})), \mathcal{D}_{\text{val}}), \tag{6}$$

where $\widehat{\mathbf{w}}$ is the trainable weight of the lightweight model. Refer to § 6 to see how our computationally efficient method decreases computation time while maintaining performance.

### 4.3 Multi-Objective Bayesian Optimization for Model Fusion

After completing the construction of the set $\mathcal{S}$ with $N$ individual models, the next stage involves selecting appropriate averaging coefficients $\boldsymbol{\delta} \in [0, 1]^N$ to ensure the enhanced metric performance of a fused model. To achieve this, we can leverage metrics $\{f_{\text{metric}}^{(k)}\}_{k=1}^{K}$ and apply a BO procedure to obtain optimal averaging coefficients $\boldsymbol{\delta}^\star$, similar to the optimization process for the hyperparameter set $\boldsymbol{\lambda}$. However, restricting the consideration to metric performance solely on $\mathcal{D}_{\text{val}}$ may result in our fused weights $\bar{\mathbf{w}}$ overfitting to $\mathcal{D}_{\text{val}}$ and exhibiting poor generalization to the true data distribution, due to the complex and sharp nature of the metric landscape which is observed in § 3.1. To tackle this challenge, when optimizing $\boldsymbol{\delta}$, we propose to minimize both $f_{\text{loss}}$ and $\{f_{\text{metric}}^{(k)}\}_{k=1}^{K}$ by employing MOBO identify a Pareto frontier defined as follows:

$$\mathcal{P} = \left\{ \boldsymbol{\delta}^\star \mid \boldsymbol{\delta}^\star = \arg\min_{\boldsymbol{\delta}} \left( l(\boldsymbol{\delta}), l_1(\boldsymbol{\delta}), \dots, l_K(\boldsymbol{\delta}) \right) \right\}, \tag{7}$$

where $l(\boldsymbol{\delta}) := \bar{f}_{\text{loss}}(\mathcal{M}(\boldsymbol{\theta}_{\text{init}}, \bar{\mathbf{w}}(\boldsymbol{\delta})), \mathcal{D}_{\text{val}})$ and $l_k(\boldsymbol{\delta}) := \bar{f}_{\text{metric}}^{(k)}(\mathcal{M}(\boldsymbol{\theta}_{\text{init}}, \bar{\mathbf{w}}(\boldsymbol{\delta})), \mathcal{D}_{\text{val}})$ for $k \in [K]$. Note that $\bar{\mathbf{w}}(\boldsymbol{\delta})$ denotes a fused set of weights with an averaging coefficient vector $\boldsymbol{\delta}$, i.e., $\bar{\mathbf{w}}(\boldsymbol{\delta}) = \sum_{i=1}^{N} \delta_i \mathbf{w}_i$ where $\mathbf{w}_i \in \mathcal{S}$ for $i \in [N]$ and $N$ is the number of models to fuse.

Here, we utilize the EHVI strategy, which is described in the work by Emmerich et al. [16]. The hypervolume, in this context, is defined as a volume size between $\mathcal{P}$ and a reference point $\mathbf{r}$. We set the reference points as a zero vector. To enhance the optimization of the hypervolume improvement objective, we employ the logarithmic form of $q$NEHVI algorithm [10, 3], which is implemented with the BoTorch framework [4]. As highlighted in § 2, this algorithm has proven effective in practical multi-objective optimization scenarios. This makes it well-suited to handle the complex and sharp nature of our metric landscape, enabling it to successfully identify the optimal $\boldsymbol{\delta}^\star$. We run MOBO for a total of $5|\boldsymbol{\delta}| = 75$ iterations to find the optimal coefficients $\boldsymbol{\delta}^\star$.

In our case, additional constraints are in place for executing MOBO, specifically 1) equality constraints and 2) inequality constraints for $\boldsymbol{\delta}$. To address the inequality constraints (i.e., $\delta_i \geq 0$), we follow the work by Gardner et al. [17] to incorporate constraints into the acquisition function. To deal with the equality constraints $\sum_{i=1}^{N} \delta_i = 1$, we simply normalize the output of the acquisition function. Refer to Algorithm 1 in Appendix B for the summary of BOMF.

## 5    Related Work

**Model Fusion for Pre-Trained Language Models.**    Due to the increasing number of model parameters in recent PLMs, there has been a significant increase in both memory requirements and

Table 1: **Results on Medium-Sized Language Models.** We conduct the text classification task using RoBERTa-base on a subset of the GLUE benchmark datasets, and the question-answering task using T5-base on the SQuAD2.0 dataset. ACC, F1, and EM denote accuracy, F1 score, and Exact Match, respectively.

| | RoBERTa-base | | | | | | T5-base |
|---|---|---|---|---|---|---|---|
| Method | RTE (Acc) | MRPC (F1) | SST-2 (Acc) | QNLI (Acc) | QQP (F1) | MNLI (Acc) | SQuAD2.0 (F1/EM) |
| Grid Fine-Tune | 77.78 | 92.39 | 94.87 | 92.62 | 88.16 | 87.41 | 78.18/72.83 |
| HPBO (Full) | 78.57 | 92.78 | 95.11 | 93.01 | 88.58 | 87.46 | 78.28/73.29 |
| SWA | 78.62 | 92.24 | 95.42 | 92.81 | 88.49 | 87.41 | 80.31/74.85 |
| OTfusion | 77.08 | 92.82 | 94.27 | 92.22 | 88.34 | 87.43 | 80.75/74.99 |
| Greedy SWA | 80.70 | 92.83 | 95.54 | 93.16 | 88.64 | 87.45 | 80.63/75.44 |
| Learned SWA | 81.40 | 92.81 | 95.31 | 92.94 | 88.38 | 87.41 | 80.65/74.23 |
| TWA | 81.23 | 91.58 | 95.54 | 93.00 | 87.85 | 87.42 | 80.29/74.79 |
| BOMF$^\dagger$ (ours) | **81.75** | 93.37 | **95.65** | **94.83** | 88.66 | 87.51 | 80.82/75.79 |
| BOMF (ours) | 81.40 | **93.90** | 95.54 | 93.50 | **88.68** | **87.86** | **81.82/76.21** |

computational costs [73, 8, 63]. Consequently, there is growing attention on a research direction aimed at enhancing the performance of PLMs while simultaneously managing computational costs and memory requirements through the exploration of model fusion methods [54, 71, 9]. However, most of these studies have focused on fusing the models fine-tuned on different tasks, aiming to develop a single multi-task learner. In the context of a single-task fine-tuning scenario within PLM, it has been observed that the previous simple weight-averaging approaches often yield marginal improvements [70, 27]; nevertheless, the exploration into the underlying rationale of this consequence remains limited. As mentioned in § 3, we find that uniform weight averaging does not always align generalization on the loss surface with the optimal point on the metric surface, primarily due to the discrepancy between loss and metric landscapes. To address this issue, we develop a single-task model fusion method based on MOBO, finding the optimal weight combination coefficients by considering both metrics and loss functions.

**Bayesian Optimization.** BO [5, 18] is a promising strategy to optimize a black-box function. In particular, if a target objective is costly in terms of function evaluations, Specifically, BO sequentially seeks solution candidates by modeling a surrogate function and maximizing an acquisition function. In the BO community, a GP [55] is often employed as a surrogate function but diverse regression models such as Bayesian neural networks [61, 38] and tree-based models [23, 30] can be used. As a choice of acquisition function, expected improvement [26] and GP upper confidence bound [62] are often considered. Importantly, BO is more effective than other existing optimization strategies such as grid search and genetic algorithms [64]. Its efficacy has been demonstrated in a wide variety of applications such as hyperparameter optimization [58], nanostructured device design [31], and chemical reaction optimization [57]. Moreover, in the deep learning context, the necessity for efficient hyperparameter tuning via BO has risen following the increasing number of hyperparameters and parameters in models [59]. Consequently, BO is applied for hyperparameter optimization in various deep learning tasks, such as image classification [28, 34] and NLP tasks [44, 7].

# 6 Experiments

In this section, we present empirical results demonstrating the effectiveness of BOMF in various NLP tasks. We compare our method to five basic algorithms aimed at finding a high-performing solution. **Grid Fine-Tune** is a simple fine-tuning method that selects the best-performing checkpoint using grid search. **HPBO** utilizes optimal hyperparameters obtained by § 4.2 for fine-tuning the baselines. **SWA** is an optimization technique that averages model parameters obtained during training. **Greedy SWA** is a modified version of SWA inspired by Greedy Soups [70], sorting weights based on metric performance on $\mathcal{D}_{val}$ and including them in $\bar{\mathcal{S}}$ only if they improve $\bar{w}$'s performance. **Learned SWA**, inspired by Learned Soup [70], learns the coefficients $\delta$ based on the loss after fine-tuning. For medium-sized language models, we tested a variant of Transformer OTfusion [24], aligning pre-trained weights before averaging. Additionally, we experimented with **TWA** [36], a recent SWA variant that reconstructs $\mathcal{S}$ by finding weight space basis vectors and learns $\delta$ based on the loss.

Table 2: **Results on Large Language Models.** We compare the results of BOMF and baseline methods using the SAMSum and KorMedMCQA datasets for summarization and medical multiple choice question-answering tasks with LLaMA2-7B and LLaMA3-8B. R1, R2, and RL denote Rouge-1, Rouge-2, and Rouge-L scores for summarization. Doctor, Nurse, and Pharm denote evaluation results for medical question answering in each respective field, using accuracy as the metric.

(a) Summarization (SAMSum)

| METHOD | R1 | R2 | RL | AVG. |
|---|---|---|---|---|
| HPBO (RANK 64) | 52.66 | 28.22 | 44.33 | 41.73 |
| SWA | 51.81 | 27.61 | 43.55 | 40.99 |
| GREEDY SWA | **53.40** | 28.06 | 43.31 | 41.49 |
| LEARNED SWA | 52.93 | **28.97** | 44.04 | 41.98 |
| BOMF† (OURS) | **53.40** | 28.78 | 44.38 | **42.19** |
| BOMF (OURS) | 53.07 | 28.61 | **44.40** | 42.03 |

(b) Korean Medical Question Answering

| METHOD | DOCTOR | NURSE | PHARM | AVG. |
|---|---|---|---|---|
| ICL | 37.89 | 50.15 | 50.00 | 46.01 |
| HPBO (RANK 64) | 43.62 | 54.64 | 51.49 | 49.92 |
| SWA | 43.96 | 54.64 | 51.97 | 50.19 |
| GREEDY SWA | 43.97 | 54.64 | 51.98 | 50.20 |
| LEARNED SWA | 44.06 | 54.94 | 52.28 | 50.43 |
| BOMF† (OURS) | 45.00 | **55.70** | **52.97** | **51.22** |
| BOMF (OURS) | **45.31** | 55.37 | 52.80 | 51.16 |

In all tables, the best performance is indicated with **boldfaced underline**, while the second-best value is represented with underline in each column. The final column 'Avg.' provides a summary of overall results for each method across various datasets or metrics. The terms 'Full' and 'Freeze' in Table 1 specify the exploration of optimal hyperparameters using either the entire model or a model with half of its weights frozen, as discussed in § 4.2. Similarly, the terms 'Rank 64' and 'Rank 4' in Table 2a denote that we use the Rank 64 or the lightweight Rank 4 version of the LoRA model for the hyperparameter search, respectively. See Appendix A for the details of downstream datasets and hyperparameter selection.

## 6.1 Empirical Analysis on Medium-Sized Language Models

We begin by evaluating the effectiveness of BOMF on medium-sized language models using RoBERTa-base [40] and T5-base [51]. For RoBERTa-base, we performed text classification tasks using the GLUE benchmark datasets [65]. For T5-base, we carried out the question-answering task with the SQuAD2.0 [52] dataset. For both models, we fine-tuned the weights directly on the downstream datasets.

Table 1 shows that BOMF consistently outperforms other baselines across all model structure and datasets.[3] Notably, the performance of HPBO, which uses hyperparameters obtained from § 4.2 with the full model, surpasses Grid Fine-Tune for most datasets. These results demonstrate that our BO-based hyperparameter search framework effectively discovers optimal hyperparameters compared to grid search. Refer to Appendix C.4 for the performance of freeze HPBO, which uses a lightweight model for hyperparameter optimization. Freeze HPBO also clearly outperforms Grid Fine-Tune which proves the effectiveness of our BO-based hyperparameter search. Also, it is evident that model fusion methods, except BOMF, lead to performance declines compared to HPBO, as discussed in § 3.1, in certain datasets. On the contrary, BOMF consistently betters the performance compared to HPBO, yielding that our method with MOBO effectively finds optimal $\delta^\star$ even in complex and sharp metric landscapes. Refer to Table 16 for the complete results.

## 6.2 Empirical Analysis on Large Language Models

We further validated the effectiveness of our proposed method by fine-tuning larger models using LoRA. Specifically, we experimented with LLaMA2-7B and LLaMA3-8B on tasks such as summarization using the SAMsum [19] dataset, Korean multi-choice medical question answering using the KorMedMCQA [32] dataset, and dialogue generation using the E2E [47] dataset. In the summarization task, while Learned SWA exhibited the best performance in terms of Rouge-2, BOMF surpassed Learned SWA in average performance across all metrics, as illustrated in Table 2a. Notably, for Rouge-L, only BOMF improved over HPBO, highlighting the effectiveness of the multi-objective approach in BOMF. Furthermore, as shown in Table 2b, our model not only outperforms other baselines but also demonstrates that fine-tuning remains essential for specific tasks despite the rise of

---

[3]A † symbol indicates results from the trajectory found using full HPBO, while results without the symbol indicate trajectories found using freeze HPBO.

in-context learning (ICL) [14]. This highlights the necessity of BOMF, which efficiently identifies hyperparameters and provides an effective fine-tuning solution through model fusion. The results for E2E can be found in Appendix C.4.

### 6.3 Ablation Study

**Number of Frozen Layers.** To analyze the efficiency of memory and compute when using a lightweight model in the BO procedure to find $\lambda^\star$, we conduct a study using RoBERTa-base on the RTE and MRPC datasets. As presented in Table 3, the use of a lightweight model successfully identifies favorable hyperparameters that yield good performance while reducing the number of parameters by up to 25% and the computation time by up to 66%. This efficiency is achieved by caching outputs from the frozen layers. By systematically freezing layers from the tail of the model, we can cache the outputs from these frozen layers and reuse them during the training process.

Table 3: **Results on the Varying Number of Frozen Layers.** Comparison of the number of parameters and relative training wall-clock time per epoch when optimizing hyperparameters across different numbers of frozen layers, using RoBERTa-base fine-tuned on the RTE and MRPC datasets.

| TASK | PARAMS | RELATIVE TIME | RTE | MRPC |
|---|---|---|---|---|
| GRID FINE-TUNE | 125M | ×1 | 77.78 | 92.39 |
| FULL | 125M | ×1 | 78.57 | 92.78 |
| FREEZE 2 | 110M | ×0.53 | 78.50 | 92.39 |
| FREEZE 4 | 96M | ×0.44 | 78.34 | 92.36 |
| FREEZE 6 | 82M | ×0.34 | 78.49 | 92.72 |

**Multiple Objectives.** To validate the efficacy of using multiple objectives when determining optimal $\delta$, we compare BOMF with single-objective baselines using T5-base on the SQuAD2.0 dataset. In this task, we consider two metrics: F1 score and Exact Match. Table 4 shows that relying on only one specific metric slightly increases the objective metric but results in a significant performance drop for the other metric. This outcome suggests

Table 4: **Comparison of Using Multi-Objective and Single-Objective Approaches.** Results of BOMF and single-objective BO baselines with T5-base fine-tuned on the SQuAD2.0 dataset.

| METRIC | F1 | EM | AVG. |
|---|---|---|---|
| F1 ONLY | 81.01 | 75.09 | 78.05 |
| EM ONLY | 80.40 | 75.87 | 78.13 |
| BOMF | 80.82 | 75.79 | 78.31 |

that using single-objective BO is appropriate when aiming to find a model optimized for a specific metric, while the use of MOBO is more suitable for discovering an optimal fused model that achieves high performance across various metrics. Refer to Appendix C.3 for further ablation studies.

## 7 Conclusion

In this paper, we empirically remarked two intriguing findings on loss and metric landscapes and hyperparameter alignment. Then, motivated by the observations mentioned above, we proposed a novel BO-based BOMF algorithm for model fusion. Our method utilizes the BO and MOBO frameworks to seek optimal fine-tuning hyperparameters and averaging coefficients, respectively. We validated that our proposed method exhibits improved performance on both NLU and NLG tasks on middle- and large-scale PLMs.

**Limitations and Future Work.** As discussed in § 3.2, compelling future research involves the theoretical analysis of the hyperparameter alignment phenomenon. Moreover, we empirically observed that when utilizing quantization-based low-rank approximation methods [11, 37], traditional uniform averaging methods and weighted averaging methods face challenges in effectively aggregating models. These challenges arise from the quantized weight values in the models that behave differently with averaging weights. Another research direction is the development of averaging methods for the quantization-based low-rank approximation methods.

## Acknowledgments and Disclosure of Funding

This work was partly supported by the National Research Foundation of Korea (NRF) grant funded by the Korea government (MSIT) (NRF-2022R1A5A708390812) and Institute of Information & communications Technology Planning & Evaluation (IITP) grant funded by the Korea government (MSIT) (No.RS-2019-II190075, Artificial Intelligence Graduate School Program (KAIST), No.2022-

0-00184, Development and Study of AI Technologies to Inexpensively Conform to Evolving Policy on Ethics, No.2022-0-00713, Meta-learning Applicable to Real-world Problems).

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

# A  Details of Experiments

Our implementation leverages key libraries, including PyTorch 2.0.1 [49], Huggingface Transformers [69], and BoTorch [4], to construct a robust framework for our experiments. These experiments are rigorously conducted on high-performance computing hardware, specifically NVIDIA RTX 3090 and NVIDIA RTX A6000 GPUs, to ensure the efficiency and scalability of our models. To further bolster the reproducibility of our results, we meticulously set and documented all experiment seeds, enabling precise replication of our experimental conditions and findings.

## A.1  Medium-Sized Language Models

Table 5: Detailed ROBERTa experimental setup.

| CATEGORY | DETAILS |
|---|---|
| **MODEL SPECIFICATIONS** | |
| ARCHITECTURE | TRANSFORMER |
| PRE-TRAINING | ROBERTA-BASE |
| OPTIMIZER | ADAMW |
| SCHEDULER | LINEAR SCHEDULER WITH WARMUP |
| WARMUP RATIO | 0.2 IF RTE OR MRPC ELSE 0.1 |
| LEARNING RATE | [1E-06, 1E-04] |
| BATCH SIZE | [8, 16] IF RTE OR MRPC ELSE [32, 64] |
| EPOCHS | 20 IF RTE ELSE 10 |
| **TASK SPECIFICATIONS** | |
| TASK NAME | CLASSIFICATION |
| DATASET | SUBSET OF GLUE BENCHMARKS. |

For the ROBERTa model, we evaluated the performance for classification and utilized a subset of the GLUE benchmark [65]. This benchmark serves as a comprehensive evaluation of a language model's overall NLU capabilities. The Recognizing Textual Entailment (RTE) task, which employs neutral and contradiction instances to assign a not-entailment label, is a binary classification task comprising 2,490 training instances and 277 validation instances. The Microsoft Research Paraphrase Corpus (MRPC) [13] consists of sentence pairs and corresponding labels. This task involves binary classification to determine whether a pair of sentences are semantically equivalent, utilizing the F1 score as the metric due to label imbalance. This dataset contains a total of 3,668 training and 408 validation instances. The Stanford Sentiment Treebank (SST-2) [60] includes movie reviews with associated positive/negative labels. The task is binary classification to discern the sentiment of a given sentence as positive or negative, with 67,349 training and 872 validation instances. The Stanford Question Answering Dataset (QNLI) [52] is a question-answering task composed of paragraph-question pairs, where one sentence in the paragraph contains the answer to the human-generated question. This dataset comprises 104,743 training and 5,463 validation instances. The Quora Question Pairs dataset (QQP) [67] involves determining whether two questions are semantically equivalent, again using the F1 score as the metric due to label imbalance, with 363,846 training and 40,430 test instances. Lastly, The Multi-Genre Natural Language Inference Corpus (MNLI) [68] is labeled for textual entailment across genre pairs, primarily consisting of premise and hypothesis sentence pairs. This task predicts the relationship between these sentences in three categories. The dataset includes 392,702 training and 9,815 validation instances, of which we used the matched case of the validation set. We conducted experiments by selecting two datasets from each GLUE benchmark based on their size scale. Additionally, specific details on the fine-tuning methods can be found in Table 5.

Table 6: Detailed T5 experimental setup.

| CATEGORY | DETAILS |
|---|---|
| **MODEL SPECIFICATIONS** | |
| ARCHITECTURE | TRANSFORMER |
| PRE-TRAINING | T5-BASE |
| OPTIMIZER | ADAMW |
| LEARNING RATE | [1E-06, 1E-04] |
| BATCH SIZE | [32, 64] |
| GRADIENT ACCUMULATION STEP | 2 |
| EPOCHS | 3.0 |
| **TASK SPECIFICATIONS** | |
| TASK NAME | Question Answering |
| INPUT TEXT | *"question: {question} context: {context}"* |
| LABEL TEXT | *"{answer}"* |
| DATASET | SQUAD2.0 |
| MAX NEW TOKENS | 10 |

For the T5-base model, we utilize the Stanford Question Answering Dataset (SQuAD 2.0) [52]. This dataset comprises 130,319 training pairs and 11,873 validation pairs of questions and answers. The dataset can be accessed through the Hugging Face datasets library.[4] Details on our fine-tuning procedures are provided in Table 6. Furthermore, we assess the generated answers by adhering to the code established in the official SQuAD 2.0 repository.[5]

## A.2 Large Language Models

In our experiments with the LLaMA2-7B[6] model, we focused on two tasks: summarization and dialogue generation. For the summarization task, we employed the Samsung Abstractive Messenger Summarization (SAMSum) dataset [19], which consists of 14,732 training samples, 818 validation samples, and 819 test samples. For the dialogue generation task, we selected the End-to-End NLG Challenge (E2E) dataset [47]. This dataset includes 42,061 training samples, 4,672 validation samples, and 4,693 test samples. Details of our fine-tuning process are provided in Table 7. Notably, in the case of the E2E dataset, the test set typically contains around five common inputs with a variety of labels. To save time, we conducted a generate process for one common input and used the different labels as multiple references to calculate the metrics. Consequently, for evaluation, the sentences generated by the model are based on a unique label, totaling 630 sentences. This accounts for the discrepancy in experimental performance between our study and that presented in the original paper of the E2E dataset [47]. All metrics including BLEU, METEOR, and ROUGE were computed using the Huggingface evaluate library.[7]

To demonstrate that fine-tuning is still necessary in specific domains and to show the effectiveness of our method in finding the best model under these circumstances, we conducted evaluations using the Korean Medical Multiple Choice Question Answering (KorMedMCQA) dataset [32]. For

---

[4] https://huggingface.co/datasets/squad_v2
[5] https://rajpurkar.github.io/SQuAD-explorer
[6] https://huggingface.co/meta-llama/Llama-2-7b-hf
[7] https://huggingface.co/evaluate

Table 7: Detailed LLaMA2-7B experimental setup.

| Category | Details |
|---|---|
| **Model Specifications** | |
| Architecture | Transformer |
| Pre-training | LLaMA-2-7B |
| LoRA alpha | 16 |
| LoRA dropout | 0.1 |
| Optimizer | AdamW |
| Learning rate | [1e-06, 1e-03] |
| Batch size | [16, 32] |
| Gradient accumulation step | 2 |
| Epochs | 2 |
| **Task Specifications** | |
| Task name | Summarization |
| Prompt | *"Summarize the following dialogue that is delimited with triple backticks."* |
| Dataset | SamSum |
| Task name | Dialogue Generation |
| Prompt | *"Generate a natural language description for the following restaurant attributes."* |
| Dataset | E2E |
| **Natural Language Generation Details** | |
| Top-p | 0.9 |
| Temperature | 1e-12 |
| Max new tokens | 100 |

batch learning, we used text segments with a maximum sequence length not exceeding 512 tokens. Consequently, the train, test, and validation sets for doctors contained 1,890, 285, and 164 examples, respectively; for nurses, the train, test, and validation sets contained 582, 291, and 291 examples; and for pharmacists, the train, test, and validation sets contained 692, 614, and 300 examples, respectively. For in-context learning, we provided examples within this length limit, and for classification fine-tuning, we used a linear head. For this, we used the LLaMA3-8Bs model, the latest multilingual open-source large language model. This version was downloaded from this link.[8] More specific details about the model and experiments can be found in Table 8.

## A.3 Bayesian Optimization

**Details of HPBO.** In the HPBO experiments, the number of iterations varied depending on the size of each dataset. Specifically, 20 iterations were conducted for the RTE dataset, while 10 iterations

---

[8] https://huggingface.co/meta-llama/Meta-Llama-3-8B-Instruct

Table 8: Detailed LLaMA3-8B experimental setup.

| Category | DETAILS |
|---|---|
| **MODEL SPECIFICATIONS** | |
| ARCHITECTURE | TRANSFORMER |
| PRE-TRAINING | LLaMA-3-8B-INSTRUCTION |
| LoRA ALPHA | 16 |
| LoRA DROPOUT | 0.0 |
| OPTIMIZER | ADAMW |
| LEARNING RATE | [1E-06, 1E-03] |
| BATCH SIZE | [8, 16] |
| GRADIENT ACCUMULATION STEP | 2 |
| EPOCHS | 10 |
| **TASK SPECIFICATIONS** | |
| TASK NAME | KOREAN MEDICAL QUESTION ANSWERING |
| PROMPT | 다음은 의사 면허 시험의 의료 질문입니다.
질문을 읽고 올바른 답을 선택하세요.

항문압 측정 검사에서 항문 압력이 증가하는 경우는?
   A. 직장질루 (RECTOVAGINAL FISTULA)
   B. 항문열창 (ANAL FISSURE)
   C. 대변실금 (FECAL INCONTINENCE)
   D. 대변메막힘 (FECAL IMPACTION)
   E. 직장탈출증 (RECTAL PROLAPSE)
답: |
| DATASET | KORMEDMCQA |

were carried out for the MRPC, SST2, and QNLI datasets. For the QQP and MNLI datasets, 8 iterations were performed. In addition, the SQuAD 2.0, SAMSum, and E2E datasets each underwent 10 iterations. These iteration counts were determined based on the respective sizes of the datasets. For single metric tasks, the chosen objective was the single valid metric itself. Conversely, for multi-metric tasks, the objective was the sum of all valid metrics.

**Details of Sampling Fusion Members.** We collected fusion members at step intervals ranging from 0.5 to 2.0 times the point of convergence identified in the training trajectory which represented $B$ in Algorithm 1, adjusting the process to yield approximately 15 members in total. Additionally, for the ROBERTa model, we employed PyTorch's official SWA scheduler with cosine annealing. For the T5 and LLaMA models, we do not use any additional scheduler for collecting SWA members.

**Details of MOBO.** In the case of MOBO, we initially provided the length of the fusion member and conducted iterations five times the total number of fusion members. This approach follows the common practice in BO of determining the initial points and the number of iterations based on the input dimension, allowing for the option to perform more iterations for improved performance. Furthermore, due to the differing scales of the loss and each metric, we applied min-max normalization

---

**Algorithm 1** Bayesian Optimization Model Fusion

---

**Require:** Training set $\mathcal{D}_{\text{trn}}$, validation set $\mathcal{D}_{\text{val}}$, initial pre-trained weights $\boldsymbol{\theta}_{\text{init}}$, initial hyperparameters $\boldsymbol{\lambda}_{\text{init}}$.

**Ensure:** Optimized hyperparameters $\boldsymbol{\lambda}^*$, combination coefficients $\boldsymbol{\delta}^*$.

1: Initialize model $\mathcal{M}$ with $\theta_{\text{init}}$, Optionally freeze layers and cache intermediate features.
2: Initialize BO with GP model, starting with $\boldsymbol{\lambda}_{\text{init}}$
   and prior data $\mathcal{H}_0 = (\boldsymbol{\lambda}_{\text{init}}, \sum_{k=1}^{K} \bar{f}_{\text{metric}}^{(k)}(\mathcal{M}(\boldsymbol{\theta}_{\text{init}}, \mathbf{w}(\boldsymbol{\lambda}_{\text{init}})), \mathcal{D}_{\text{val}}))$.
3: **for** $i = 1$ **to** $I$ iter **do**
4:     Define LogEI using current GP.
5:     Find $\boldsymbol{\lambda}_i$ by optimizing LogEI.
6:     Training $\mathcal{M}(\boldsymbol{\theta}_{\text{init}}, \mathbf{w})$ with $(\mathcal{D}_{\text{trn}}, \boldsymbol{\lambda}_i)$.
7:     Evaluate $\sum_{k=1}^{K} \bar{f}_{\text{metric}}^{(k)}(\mathcal{M}(\boldsymbol{\theta}_{\text{init}}, \mathbf{w}(\boldsymbol{\lambda}_i)), \mathcal{D}_{\text{val}})$.
8:     Update GP model with new data $(\boldsymbol{\lambda}_i, \sum_{k=1}^{K} \bar{f}_{\text{metric}}^{(k)}(\mathcal{M}(\boldsymbol{\theta}_i, \mathbf{w}(\boldsymbol{\lambda}_i)), \mathcal{D}_{\text{val}}))$.
9: **end for**
10: Collect $\boldsymbol{\lambda}^* = \arg\min_{\lambda} \sum_{k=1}^{K} \bar{f}_{\text{metric}}^{(k)}(\mathcal{M}(\boldsymbol{\theta}_{\text{init}}, \mathbf{w}(\boldsymbol{\lambda})), \mathcal{D}_{\text{val}})$
11: $B^* \leftarrow \arg\min_B \bar{f}_{\text{metric}}^{(k)}(\mathcal{M}(\boldsymbol{\theta}_{\text{init}}, \mathbf{w}_B(\boldsymbol{\lambda}^*)), \mathcal{D}_{\text{val}})$
12: $\mathcal{S} \leftarrow \{\}$.
13: **for** $j = 1$ **to** $J$ step **do**
14:     Optimize $\mathcal{M}(\boldsymbol{\theta}_{\text{init}}, \mathbf{w}_j)$ with $(\mathcal{D}_{\text{trn}}, \boldsymbol{\lambda}^*)$.
15:     **if** $j \geq 0.5B^*$ **then**
16:         $\mathcal{S} \leftarrow \mathcal{S} \cup \mathcal{M}(\boldsymbol{\theta}_{\text{init}}, \mathbf{w}_j)$
17:     **end if**
18: **end for**
19: Set reference point $\mathbf{r}$.
20: $\mathcal{L}_{\text{init}} \leftarrow \{\bar{f}_{\text{loss}}(\mathcal{M}(\boldsymbol{\theta}_{\text{init}}, \bar{\mathbf{w}}(\boldsymbol{\delta}_{\text{init}})), \mathcal{D}_{\text{val}}), \bar{f}_{\text{metric}}^{(1)}(\mathcal{M}(\boldsymbol{\theta}_{\text{init}}, \bar{\mathbf{w}}(\boldsymbol{\delta}_{\text{init}})), \mathcal{D}_{\text{val}}), \cdots$
    $, \bar{f}_{\text{metric}}^{(K)}(\mathcal{M}(\boldsymbol{\theta}_{\text{init}}, \bar{\mathbf{w}}(\boldsymbol{\delta}_{\text{init}})), \mathcal{D}_{\text{val}})\}$
21: Initialize MOBO with GP models, starting with $\bar{\mathbf{w}}_{\text{init}}$ and prior data $\mathcal{H}_0 = (\bar{\mathbf{w}}_{\text{init}}, \boldsymbol{\delta}_{\text{init}}, \mathcal{L}_{\text{init}})$.
22: Compute initial Pareto optimal set $\mathcal{P}_0$ using $\mathcal{H}_0$.
23: **for** $m = 1$ **to** $M$ iter **do**
24:     Define EHVI using current GPs.
25:     Find $\boldsymbol{\delta}_m$ by optimizing EHVI. Equation 4
26:     $\mathcal{L} \leftarrow \{\bar{f}_{\text{loss}}(\mathcal{M}(\boldsymbol{\theta}_{\text{init}}, \bar{\mathbf{w}}(\boldsymbol{\delta}_m)), \mathcal{D}_{\text{val}}), \bar{f}_{\text{metric}}^{(1)}(\mathcal{M}(\boldsymbol{\theta}_{\text{init}}, \bar{\mathbf{w}}(\boldsymbol{\delta}_m)), \mathcal{D}_{\text{val}}), \cdots$
    $, \bar{f}_{\text{metric}}^{(K)}(\mathcal{M}(\boldsymbol{\theta}_{\text{init}}, \bar{\mathbf{w}}(\boldsymbol{\delta}_m)), \mathcal{D}_{\text{val}})\}$
27:     Update GPs with new data $(\boldsymbol{\delta}_m, \mathcal{L})$
28:     Update $\mathcal{P}_m$.
29: **end for**
30: Collect $\boldsymbol{\delta}^* = \arg\max_{\boldsymbol{\delta}} \text{EHVI}(\boldsymbol{\delta}; \mathcal{L}, \mathbf{r})$

---

to adjust the scales, utilizing the lowest value of single model performance obtained after the convergence point in the trajectory collection of members, rounded to the nearest value, as the minimum. The maximum values were determined by adding 0.1 for metric and 1.0 for loss respectively to this minimum value for use as the maximum. If there is a more critical metric or criterion, one can freely modify the optimization by placing weights according to the user's intent. No additional weights were applied in our experiments.

# B   Proposed Algorithm

In this section, we provide an overview of the complete process of BOMF encapsulated in the algorithm. Algorithm 1 systematically incorporates all three steps: 1) hyperparameter search via BO as detailed in § 4.2, 2) fusion member sampling as outlined in § 4.1, and 3) identification of optimal $\boldsymbol{\delta}^\star$ and model fusion through MOBO in § 4.3.

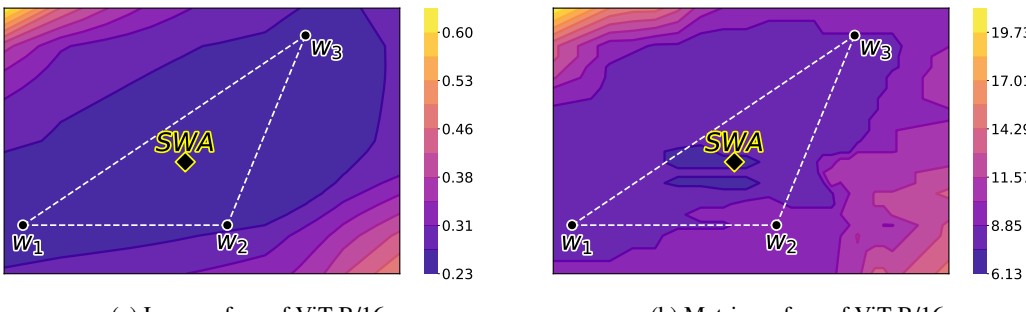

(a) Loss surface of ViT-B/16.        (b) Metric surface of ViT-B/16.

Figure 4: Visualization of the loss landscape over parameters (a) and the metric landscape over parameters (b) for the vision task. The metric is accuracy error. We fine-tune the ImageNet-21k pre-trained ViT-B/16 [15] model on the Caltech-101 dataset. The members of the SWA for each figure are denoted as $w_1, w_2, w_3$. Here we can see similar trends with the ResNet-50 case.

Table 9: Spearman's rank correlation coefficient value of (a) ResNet18 model on the CIFAR10 dataset, (b) ViT-B/16 model on the Caltech101 dataset, (c) ROBERTa-base model on the SST-2 dataset, (d) T5-base model on the SQuAD2.0 dataset, and (e) LLaMA2-7B model on the E2E dataset. Here we used 15 fine-tuned weights for each task to measure the Spearman's rank correlation. A higher value indicates a higher correlation between metric and loss value.

| METRIC | (A) | (B) | (C) | (D) | (E) |
|---|---|---|---|---|---|
| SPEARMAN'S RCC | 0.6182 | 0.6558 | 0.1430 | 0.2150 | 0.2286 |

## C   Additional Experiments

In this section we demonstrate additional experiments not included in the main article.

### C.1   Additional Experiments on Loss and Metric Landscapes

We also explored the potential loss metric discrepancy in PLMs, which might originate from the inherent features of transformer attention or from the use of adaptive optimizers. To analyze deeper, we visualize the loss-metric surface of the ViT-B/16 model per-trained on ImageNet-21k, using Adam optimizer which is the same as the optimizer of our language model. According to Figure 4, unlike in PLMs, the optimal points for loss and performance metrics in the ViT-B/16 model were aligned, indicating a distinct behavior between language and vision transformers in this context. Additionally, we have also undertaken the measurement of Spearman's rank correlation between loss and metric values across a range of models and tasks in both NLP and CV domains. Table 9 clearly shows that the correlations in NLP tasks are less than in CV tasks.

### C.2   Additional Experiments on Hyperparameter Alignment

In our experiments utilizing LORA, we aimed to verify whether the optimal hyperparameters align when using a smaller rank to reduce costs, compared to using a larger rank. Figure 5 demonstrates that the optimal batch size and learning rate align even when the number of LoRA ranks varies. In addition to these hyperparameters, we investigated the potential alignment of the beta parameter of the Adam optimizer, which is the standard optimizer for training large language models, as well as the scheduler.

Figure 6 and Figure 7 indicate that, with the exception of the scenario where all layers are frozen, the optimal beta parameter of the Adam optimizer consistently aligns regardless of the number of frozen layers or the LoRA rank. Figure 9 and Figure 10 suggest that the optimal points of different learning rate schedules can be aligned according to the number of frozen layers.

These results indicate that by employing a lightweight model, we can identify the optimal hyperparameters, thereby simplifying the hyperparameter optimization process and reducing computational time.

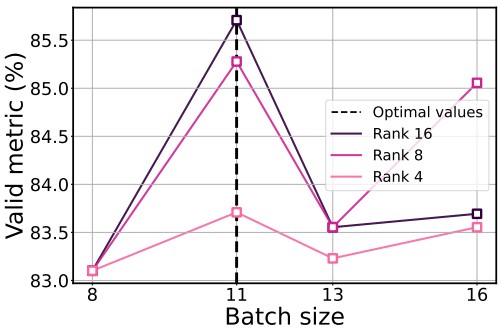 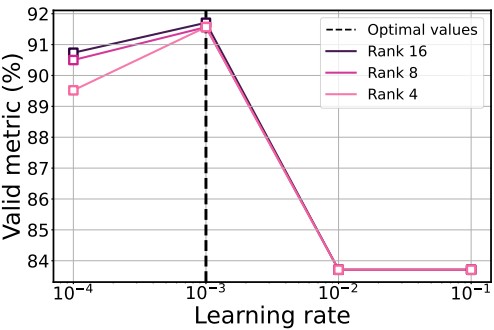

(a) Validation metric results for the varying batch size and the number of LoRA rank.

(b) Validatoin metric results for the varying learning rate and the number of LoRA rank.

Figure 5: Validation loss and metric (F1 score) results for the varying hyperparameter ((a) batch size, (b) learning rate) and the number of LORA rank for the ROBERTa on MRPC dataset. (a) and (b) indicate that the optimal hyperparameters consistently align well across different numbers of LORA rank.

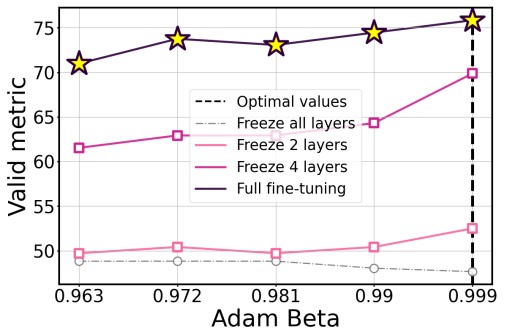 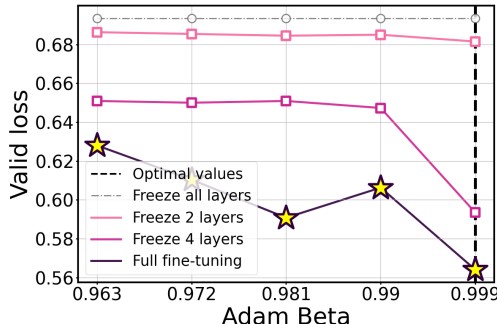

(a) Validation metric (Accuracy) for varying beta parameters and frozen layers.

(b) Validation loss for varying beta parameters and frozen layers.

Figure 6: Results for the ROBERTa-base model on the RTE dataset. (a) and (b) indicate that the optimal hyperparameters align well across different numbers of frozen layers, except when all pre-trained layers are frozen.

## C.3    Additional Ablation Experiments

**Metric Function in Multi-Objective Bayesian Optimization.**    In the § 3, we pointed out the issues of existing methods that perform fusion based solely on loss, particularly due to the discrepancy between metric and loss in language models. We demonstrated that our method outperforms the learned-SWA method, which relies only on loss. To strengthen our claim, we additionally compared the loss and metric values between optimization processes with and without metrics. Specifically, we examined the loss and metric values of weights along the training trajectory between the initial point and the optimized point of Learned SWA and BOMF. Figures 9 and 10 show that BOMF generates complex trajectories for both loss and metric, exploring solutions based on both criteria. In contrast, Learned SWA, relying solely on the loss function, gets trapped in local minima around the starting point, failing to discover the optimal solution. This suggests that BOMF's exploration property and the inclusion of metrics help escape local minima and discover more robust solutions.

**Loss Function in Multi-Objective Bayesian Optimization.**    BOMF adopts both loss and metric as objectives when optimizing $\delta$. This approach is based on the understanding that the loss provides a macroscopic guide for overall metric performance. As seen in Figure 1, the solution of SWA is optimal on the loss surface but not on the metric surface. In contrast, BOMF performs well on both the loss and metric surfaces. These results indicate that BOMF ensures a high correlation between loss and metric values, preventing overfitting to either loss or metric during validation and enhancing the robustness

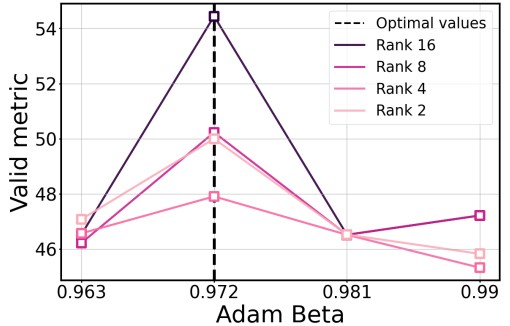
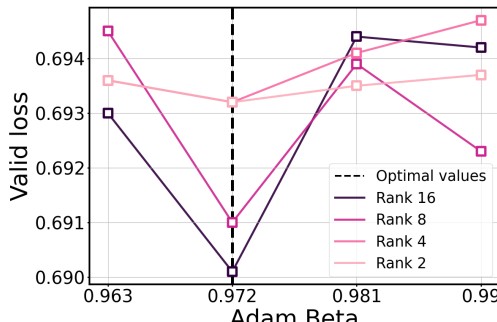

(a) Validation metric (Accuracy) for varying beta parameters and LoRA ranks.

(b) Validation loss for varying beta parameters and LoRA ranks.

Figure 7: Results for the ROBERTa-base model on the RTE dataset. (a) and (b) highlight that optimal hyperparameters align well across different numbers of LoRA ranks, emphasizing the importance of parameter tuning.

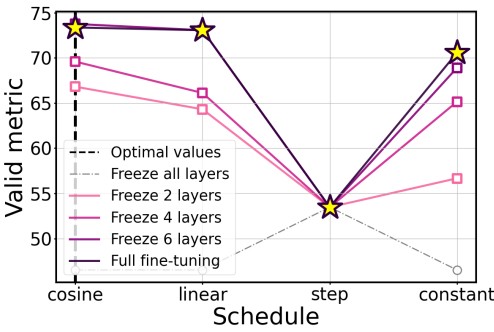
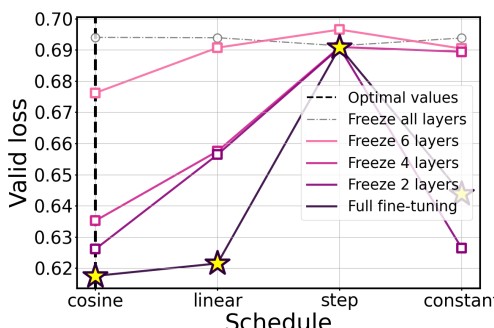

(a) Validation metric (Accuracy) for varying learning rate schedule methods and frozen layers.

(b) Validation loss for varying learning rate schedule methods and frozen layers.

Figure 8: Results for the ROBERTa-base model on the RTE dataset. (a) and (b) demonstrate that optimal hyperparameters are consistent across different numbers of frozen layers, indicating the critical role of hyperparameter choices.

of the weights on the test dataset. This clearly shows that including the loss function provides more useful guidance than optimizing $\delta$ exclusively in a complex and sharp metric landscape.

Moreover, as demonstrated in Tables 10 and 11, our approach, which utilizes both loss and metric, improves the correlation between them. This leads to reaching the point of optimal performance, as shown in Tables 14, 15 and 17. These findings support that BOMF outperforms Bayesian optimization using a single metric without the loss function.

**Multiple Metric Functions in Multi-Objective Bayesian Optimization.** For tasks with multiple metrics, we optimized using all the available metrics. Therefore, it is important to investigate how multiple metrics impact the optimization process. In Table 4, we validated this through performance evaluations, but we also assessed changes in correlation. Table 11 shows that optimizing with a single metric can weaken the correlation between the loss and other metrics.

**Impact of the Better Trajectories in Model Fusion Performance** BOMF aims to construct the best-performing single model through model fusion within a parameter space. As detailed in § 4.1, different combinations of fine-tuning hyperparameters (such as learning rate and batch size) yield varied generalization performances after the fine-tuning process. Therefore, to identify the best-performing single model after the model fusion, we must first determine the optimal hyperparameters that yield the best-performing single model before the fusion. In Table 14, we can confirm that the

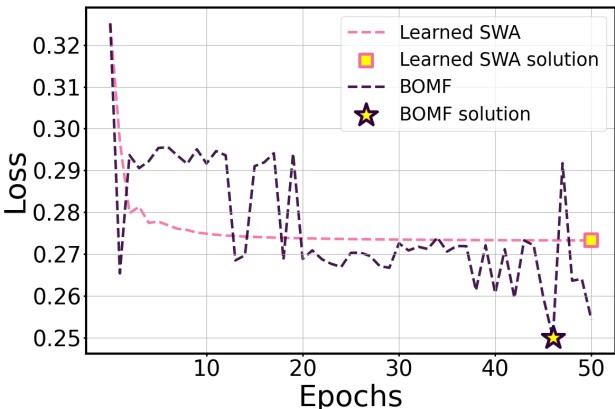

Figure 9: Visualization of the evolution of loss throughout the optimization phases. We conduct experiments on the RTE dataset with the ROBERTa-base model. Learned SWA, an optimization process without metrics, tends to converge to local minima close to the starting point. Conversely, BOMF exhibits a more successful exploration, incorporating metrics, and ultimately discovers a more robust solution.

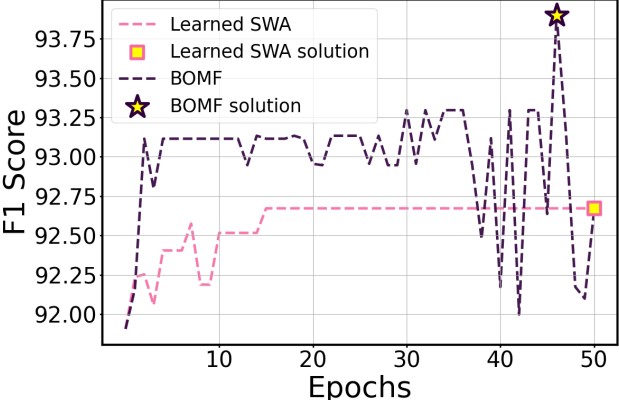

Figure 10: Visualization of the evolution of metric throughout the optimization phases. We conduct experiments on the RTE dataset with the ROBERTa-base model. Like the loss, Learned SWA tends to converge towards local minima without enhancing metric performance after the convergence. In contrast, BOMF effectively navigates through explorations to discover a high-performing solution.

performance of the final BOMF, executed after finding good hyperparameters through outerbo, is better than the fusion performance executed with hyperparameters obtained from grid search.

**Performance of Conventional Optimization Strategies for Each Task.** Additionally, we conducted experiments to directly compare the original optimizers with BOMF. We found the optimal hyperparameters for the original optimizers using BO and then performed experiments on subsets of the GLUE dataset using the ROBERTa model and the SAMSum dataset using the LLaMA2-7B model. The results presented in Table 12 clearly demonstrate that BOMF outperforms the best performance achievable with any of the original optimizers.

**Additional Experiments with BOMF Using LLM-Based Evaluation** To validate BOMF's performance under diverse evaluation metrics, we conducted experiments using a ChatGPT-3.5-Turbo-based approach. This method involves scoring by asking the LLM to assess the similarity between generated responses and ground-truth answers. Using this, BOMF was benchmarked against other models. As shown in Table 13, BOMF consistently outperformed these baselines, highlighting its capability to adapt not only to traditional metrics but also to newer evaluation techniques.

Table 10: Spearman's rank correlation coefficient value between loss and metric for the various optimization processes in the middle scale tasks. Here, we assess the correlation of the ROBERTa-base model on the MRPC dataset (a) and the RTE dataset (b). Additionally, we examine the correlation between loss and metric of the T5-base model on the SQuAD2.0 dataset using the F1 score (c) and the EM (d) metrics. We conduct evaluations on a total of 100 sampled sets for (a) and (b), and 30 for (c) and (d). Here, Loss BO SWA and Metric BO SWA denote the approach where we exclusively employ either the loss function or the metric during the MOBO process, respectively.

| METRIC | (A) | (B) | (C) | (D) |
|---|---|---|---|---|
| BASELINE (HPBO) | 0.1640 | 0.3256 | 0.2076 | 0.3021 |
| LOSS BO SWA | 0.1903 | 0.4334 | 0.4210 | 0.4710 |
| METRIC BO SWA | 0.2465 | 0.3755 | 0.5333 | 0.5000 |
| BOMF | 0.5189 | 0.5886 | 0.6500 | 0.6000 |

Table 11: Spearman's rank correlation coefficient value between loss and metric for the various optimization processes in large-scale tasks. We evaluate the LLaMA2-7B on the SAMSum dataset. Here, Loss BO SWA and Metric BO SWA denote the approach where we exclusively employ either the loss function or the metric during the MOBO process, respectively.

| METRIC | R1 | R2 | RL | AVG. |
|---|---|---|---|---|
| BASELINE (HPBO) | 0.4024 | 0.2000 | 0.1030 | 0.2351 |
| LOSS BO SWA | 0.5335 | 0.1306 | 0.1197 | 0.2613 |
| METRIC BO SWA | 0.5330 | 0.6060 | 0.6060 | 0.5817 |
| BOMF | 0.6445 | 0.6152 | 0.6137 | 0.6245 |

Table 12: **Results of BOMF and Other Neural Network Optimization Strategies on Various Datasets.** A higher value is better for all the metrics.

| | GLUE (ROBERTA-BASE) | | | SAMSUM (LLAMA2-7B) | | |
|---|---|---|---|---|---|---|
| OPTIMIZATION STRATEGY | RTE (ACC) | MRPC (F1) | SST2 (ACC) | R1 | R2 | RL |
| SGD | 54.37 | 88.90 | 90.11 | 50.63 | 19.34 | 41.24 |
| ADAMW | 78.49 | 92.72 | 94.41 | 52.25 | 27.52 | 44.03 |
| SWA | 76.70 | 91.73 | 94.75 | 52.21 | 27.58 | 44.04 |
| SAM | 77.34 | 92.56 | 94.77 | 52.75 | 28.55 | 44.16 |
| BOMF | **81.40** | **93.90** | **95.54** | **53.07** | **28.61** | **44.40** |

This robustness is further reinforced by BOMF's design, which combines loss with multiple metrics, enhancing its generalizability across unseen metrics. Notably, BOMF leverages BO for tuning combination coefficients, optimizing based on evaluation values rather than backward processing through the metrics themselves. This approach allows BOMF to efficiently optimize coefficients across complex evaluation settings, as shown in the ChatGPT BOMF column in Table 13.

These findings underscore BOMF's adaptability, demonstrating its resilience across varied evaluation frameworks, including those generated by LLMs.

## C.4 Full Experimental Results

In this section, we present full experimental results encompassing text classification tasks for the Masked Language Model (MLM) and question answering, summarization, and dialogue generation tasks for the autoregressive LLM. In all tables, the best performance is indicated with **boldfaced underline**, and the second-best value is represented with underline for methods that use the same best hyperparameters by § 4.2. The final column 'Avg.' provides a summary of overall results for each method across various datasets or metrics. The terms 'Full' and 'Freeze' in Tables 14 and 15 specify the exploration of optimal hyperparameters using either the entire model or a model with half of its weights frozen, as discussed in § 4.2. Similarly, the terms 'Rank 64' and 'Rank 4' in Tables 16 and 17 denote that we use the Rank 64 or the lightweight Rank 4 version of LORA model for the hyperparameter search, respectively.

Table 13: **Evaluation Results Using ChatGPT-3.5-Turbo.** (a) Baseline, (b) SWA, (c) Greedy SWA, (d) Learned SWA, (e) BOMF, and (f) ChatGPT BOMF.

| EVALUATION PROMPT EXAMPLE |
|---|
| YOU ARE AN AUTOMATED GRADING ASSISTANT HELPING A TEACHER GRADE STUDENT ANSWERS. |

THE CORRECT SUMMARY FOR THIS TEXT IS: <GROUND-TRUTH>
A STUDENT SUBMITTED THE SUMMARY: <PREDICTION>

THE STUDENT'S SUMMARY MUST BE CORRECT AND SPECIFIC BUT NOT OVERCOMPLETE. SMALL DIFFERENCES IN FORMATTING SHOULD NOT BE PENALIZED. ON A SCALE FROM 0 TO 100, WHERE 0 MEANS COMPLETELY INCORRECT AND 100 MEANS COMPLETELY CORRECT, HOW SIMILAR IS THE STUDENT'S SUMMARY TO THE GROUND TRUTH? PLEASE PROVIDE ONLY A NUMERICAL SCORE WITHOUT ANY EXPLANATION.

| | **(A)** | **(B)** | **(C)** | **(D)** | **(E)** | **(F)** |
|---|---|---|---|---|---|---|
| **GRADE** | 70.74 | 71.92 | 72.05 | 71.80 | 72.64 | 73.18 |

Table 14: **Full Results on Text Classification Task Using RoBERTa-base.** Results of BOMF and baseline methods with GLUE benchmark datasets. ACC and F1 denote metrics for each dataset, representing accuracy and F1 score, respectively.

| | DATASET | | | | | | AVG. |
|---|---|---|---|---|---|---|---|
| METHOD | RTE (ACC) | MRPC (F1) | SST-2 (ACC) | QNLI (ACC) | QQP (F1) | MNLI (ACC) | |
| GRID FINE-TUNE | 77.78 | 92.39 | 94.87 | 92.62 | 88.16 | 87.41 | 88.93 |
| GRID SEARCH INNER BO | 79.66 | 92.48 | 95.31 | 92.79 | 88.24 | 87.54 | |
| HPBO (FULL) | 78.57 | 92.78 | 95.11 | 93.01 | 88.58 | 87.46 | 89.25 |
| SWA | 78.62 | 92.24 | 95.42 | 92.81 | 88.49 | 87.41 | 89.17 |
| GREEDY SWA | 80.70 | 92.83 | 95.54 | 93.16 | 88.64 | 87.45 | 89.72 |
| LEARNED SWA | 81.40 | 92.81 | 95.31 | 92.94 | 88.38 | 87.41 | 89.71 |
| TWA | 81.23 | 91.58 | 95.54 | 93.00 | 87.85 | 87.42 | 89.44 |
| BOMF (OURS) | **81.75** | **93.37** | **95.65** | **94.83** | **88.66** | **87.51** | **90.30** |
| HPBO (FREEZE) | 78.49 | 92.72 | 94.41 | 92.71 | 88.04 | 87.45 | 88.97 |
| SWA | 76.70 | 91.73 | 94.75 | 93.21 | 88.35 | 87.44 | 88.70 |
| GREEDY SWA | 80.01 | 93.38 | 95.20 | 93.30 | 88.67 | 87.84 | 89.73 |
| LEARNED SWA | 78.79 | 93.56 | 95.20 | 93.03 | 88.43 | 87.55 | 89.43 |
| TWA | 78.29 | 92.16 | 94.87 | 92.86 | 88.59 | 87.44 | 89.03 |
| BOMF (OURS) | **81.40** | **93.90** | **95.54** | **93.50** | **88.68** | **87.86** | **90.15** |

**Text Classification.** Table 14 demonstrates the consistently better performance of BOMF over other baselines that employ the same best hyperparameters. These findings affirm the effectiveness of BOMF in the context of single-metric NLP tasks.

**Question Answering.** Table 15 presents the complete experimental results for the question-answering task. BOMF consistently surpasses other baselines utilizing the same best hyperparameters. These outcomes prove the effectiveness of BOMF in the realm of multi-metric NLP tasks, improving both F1 and EM metrics concurrently.

**Summarization.** Table 16 provides empirical evidence that BOMF achieves the highest average performance across evaluated metrics. While Learned SWA and Greedy SWA exhibit the best performance results in R1 and R2 metrics, respectively, they experience declines across other metrics. However, our approach demonstrates a consistent improvement across all metrics. These results prove the efficacy of BOMF in multi-metric NLP tasks.

**Dialogue Generation.** Table 17 provides full experimental results for the dialogue generation task. The results for BOMF showcase the highest scores for all metrics in both cases of rank 64 and rank 4. Despite the conflicting correlations between BLEU and METEOR metrics [29, 1], BOMF achieves comprehensive improvements across all metrics, distinguishing itself from other baselines that fail to

Table 15: **Full Results on Question-Answering Task with T5-base.** Results of BOMF and baseline methods with SQuAD2.0 dataset. F1 and EM denote the F1 score and Exact Match, respectively.

| METHOD | F1 | EM | AVG. |
|---|---|---|---|
| HPBO (FULL) | 78.28 | 73.29 | 75.79 |
| SWA | 80.31 | 74.85 | 77.58 |
| GREEDY SWA | 80.63 | 75.44 | 78.03 |
| LEARNED SWA | 80.65 | 74.23 | 77.44 |
| TWA | 80.29 | 74.79 | 77.54 |
| BOMF (OURS) | **80.82** | **75.79** | **78.31** |
| HPBO (FREEZE) | 78.19 | 73.43 | 75.81 |
| SWA | 81.21 | 75.63 | 78.42 |
| GREEDY SWA | 81.73 | 76.20 | 78.97 |
| LEARNED SWA | 81.24 | 75.65 | 78.45 |
| TWA | 81.21 | 75.61 | 78.41 |
| BOMF (OURS) | **81.82** | **76.21** | **79.01** |

Table 16: **Full Results on Summarization Task with LLaMA2-7B.** Results of BOMF and baseline methods with SAMSum dataset. R1, R2, and RL denote Rouge-1, Rouge-2, and Rouge-L, respectively.

| METHOD | R1 | R2 | RL | AVG. |
|---|---|---|---|---|
| HPBO (RANK 64) | 52.66 | 28.22 | 44.33 | 41.73 |
| SWA | 51.81 | 27.61 | 43.55 | 40.99 |
| GREEDY SWA | **53.40** | 28.06 | 43.31 | 41.49 |
| LEARNED SWA | 52.93 | **28.97** | 44.04 | 41.98 |
| BOMF (OURS) | **53.40** | 28.78 | **44.38** | **42.19** |
| HPBO (RANK 4) | 52.25 | 27.52 | 44.03 | 41.27 |
| SWA | 52.21 | 27.58 | 44.04 | 41.28 |
| GREEDY SWA | 53.06 | **28.99** | 44.03 | **42.03** |
| LEARNED SWA | 52.10 | 26.76 | 44.13 | 41.00 |
| BOMF (OURS) | **53.07** | 28.61 | **44.40** | **42.03** |

Table 17: **Full Results on Dialogue Generation Task with LLaMA2-7B.** Results of BOMF and baseline methods with E2E dataset. B, M, and RL denote BLEU, METEOR, and Rouge-L, respectively.

| METRIC | B | M | RL | AVG. |
|---|---|---|---|---|
| HPBO (RANK 64) | 63.09 | 80.17 | 65.71 | 69.66 |
| SWA | 62.95 | 80.07 | 65.53 | 69.52 |
| GREEDY SWA | 63.86 | 80.00 | 66.63 | 70.16 |
| LEARNED SWA | 63.65 | 79.76 | 66.34 | 69.92 |
| BOMF (OURS) | **64.70** | **80.91** | **67.53** | **71.05** |
| HPBO (RANK 4) | 63.46 | 81.26 | 67.09 | 70.60 |
| SWA | 63.50 | 81.02 | 67.08 | 70.53 |
| GREEDY SWA | 63.27 | 81.17 | 67.68 | 70.71 |
| LEARNED SWA | 64.42 | 79.04 | 66.09 | 69.85 |
| BOMF (OURS) | **64.81** | **81.28** | **67.70** | **71.26** |

achieve such comprehensive enhancements. This proves the efficacy of our multi-objective method in effectively considering multiple metrics with conflicting correlations.

## D  Societal impact

BOMF does not directly have any positive or negative societal impacts since our algorithm is for fine-tuning and model fusion. However, in the sense of developing LLMs, we can argue the societal impacts of our work. On the positive side, our work can improve the productivity of human beings, e.g., reduction of repeating tasks, and discover new scientific knowledge, e.g., artificial intelligence for scientific discovery. On the other hand, as negative societal impacts, fine-tuning LLMs on downstream tasks can still consume significant compute resources, which leads to climate change. Moreover, since our fine-tuning process aims to optimize specific metrics, there can be a potential risk of optimizing towards malicious metrics such as aggressiveness and violence. Therefore, we should be aware of potential unethical outcomes and consider responsibility in selecting and optimizing these metrics.

## E  Safeguards

We use publicly available benchmarks and open-source models widely recognized in the LLM research community. Additionally, we do not release proprietary or new datasets or models that could cause the risk of misuse. Although our work potentially has a possibility to undertake inherent misuse that is derived from public benchmarks and open-source models, we think that our method itself does not pose a high risk of misuse.

