# OpenReview forum: "Model Fusion through Bayesian Optimization in Language Model Fine-Tuning"
_NeurIPS.cc/2024/Conference — NeurIPS 2024 spotlight_

### Official Review · Reviewer_JjKU · 2024-07-06

**Soundness:** 2
**Presentation:** 3
**Contribution:** 3
**Rating:** 7
**Confidence:** 3

**Summary:**

The main contributions
- Using Bayesian Optimization for hyperparameter search on LoRA’s for full-model fine-tuning
- Using EVHI to learn coefficients to fuse models from within a checkpoint

Other contributions
- Finding discrepancy between metric and loss

**Strengths:**

- Paper is well-written and easy to follow
- Bayesian method for model fusion is interesting

**Weaknesses:**

- Empirical observations of discrepancy between metric and loss landscape are on one specific outdated model and not sure how accuracy is computed (but could be an outdated way). Same for hyperparameter alignment claim. See questions below
- Extra compute required for BOMF
- Performance improvements of BOMF are very small

**Questions:**

Major
- How much extra time is required due to the sequential nature of HBPO compared to grid search?
- Some discussion of the extra compute required for BOMF due to the extra evaluations. Does 75 iterations mean 75 evaluations are needed?
- How is accuracy computed on RoBERTa? With a classifier head on top of the representations? Nowadays, for NLP, all tasks are cast as text-to-text and accuracy is done by comparing the log probs of the tokens corresponding to the different labels.
- The claim that the best hyperparameters for fine-tuning LoRA can transfer to the full model are not that interesting for RoBERTa. Since RoBERTa is a pretty small model overall, LoRA is generally not needed and full-model fine-tuning is needed. LoRA is usually used for larger models, so showing this for the LLAMA-2 experiments would be interesting.

Minor
- Can you an algorithms description of MOBO in the main paper? Specifically, how is qNEHVI done?
- Where is freeze in Table 1 and rank 4 in Table 2? (Mentioned in line 298/315)
- What are the two variants of BOMF in Table 1 and 2?
- Why are there are no grid fine-tune results in Table 2?

---

> ### Author Rebuttal · Authors · 2024-08-07
>
> We appreciate your constructive feedback. We have answered your questions and concerns in this response. Please let us know if you have any follow-up questions.
>
> **[W1, Q3, Q4] Empirical observations of discrepancy between metric and loss landscape, and hyperparameter alignment are performed on the RoBERTa model. Thus it requires results on Llama models. Also how does the accuracy computed on RoBERTa?**
>
> First, we want to clarify that we use a classifier head for the classification task because our task involves multiple-choice questions, allowing us to directly output an answer and measure accuracy. However, we want to emphasize that BOMF is not limited to this specific scenario. It can be applied to any model as long as the loss and metrics can be calculated. Since fusion does not require backpropagation, BOMF is equally applicable in situations where there is no classifier head. And in our paper, we demonstrated that both full model fine-tuning and LoRA fine-tuning exhibit 1) metric and loss misalignment and 2) optimal hyperparameters alignment even when freeze layers or ranks change. However, we agree that extending these experiments to larger-scale models would strengthen our findings. Therefore, we conducted additional experiments on the Llama-2 model to verify if 1) and 2) still hold.
> Firstly, regarding point 2), the alignment of hyperparameters can be easily observed in Figures A.1. For point 1), the misalignment between loss and metrics is evident in Figure A.2. To further show the misalignment between loss and metrics, as well as among different metrics, we measured Spearman’s correlation using the loss and metrics of 20 sampled models. Also, we empirically prove that BOMF significantly improves this correlation by considering both loss and metrics simultaneously. Refer to Tables 9, 10, and 11 in Appendix C for full experimental results. We have included a portion of these results here. According to Table R.2 (c), (d), and (e), language tasks involving the Llama model show significantly lower correlation compared to vision tasks (a) and (b). Then, Table R.3 shows that BOMF successfully increases the correlation between loss and metrics compared to other baseline methods.
>
> **[W2, Q1, Q2] How much extra time is required for BOMF.**
>
> Please refer to our global response for answers.
>
> **[W3] Performance improvements of BOMF are very small**
>
> The improvement of BOMF is not negligible compared to other baselines. First, we would like to emphasize that our method consistently outperforms various baselines across a wide range of task types (i.e., classification, question answering, and the medical domain) and models (i.e., T5, LLaMA3). While the degree of improvement might appear marginal in some cases, it is essential to recognize that the extent of improvement, whether small or large, is relative. The fact that our method consistently achieves superior performance in nearly every case serves as an absolute metric that is universally recognizable. Our method consistently outperforms the baselines in these diverse scenarios. Specifically, BOMF achieved a 13%, 14%, 10% error improvement in the GLUE tasks, SQuAD task and KorMedQA tasks, respectively.
>
> Furthermore, the recent fusion research [1,2,3] in various scenarios has shown similar tendency to our results, showing that even if the improvement in specific tasks seems marginal, they exhibit overall superior performance compared to the baseline, proving the necessity of such methods. In particular, even when compared to these methods, our method demonstrates improved performance in almost all experimental results, indicating its effectiveness.
>
> **[Q5] Can you provide an algorithm description of MOBO in the main paper? Specifically, how is qNEHVI done?**
>
> We have already described the details of our algorithm in Sec A.1.3, B, and Algorithm 1 of the appendices. Due to the limited space of the NeurIPS submission, they should be located in the appendices. We will reorganize our contents in the final version.
>
> As described in our manuscript, qNEHVI sequentially samples optimum candidates by maximizing the expected hypervolume improvement, which can be considered as an acquisition function in Bayesian optimization. At every iteration of MOBO, it chooses the maximizer of the alternative objective rather than one of multiple unknown target objectives. We will clarify it in the final version.
>
> **[Q6, Q7] Where is freeze in Table 1 and rank 4 in Table 2?**
>
> Thank you for pointing out the confusion. The dagger symbol ($\dagger$) in Tables 1 and 2 indicates whether low rank or freeze layers were used. Specifically, "BOMF with dagger symbol ($\dagger$)" refers to cases where the best hyperparameters were determined using full rank or full layer settings, and these hyperparameters were then used for training before performing fusion. On the other hand, "BOMF without a dagger symbol" refers to cases where low rank or freeze layers were used to find the best hyperparameters. This information is clarified in the footnote on Page 8 of the paper.
>
> **[Q8] Why are there no grid fine-tune results in Table 2?**
>
> The models used in Table 2 are LLMs, so even when tuning with LoRA, the forward path takes a considerable amount of time. This made it time-prohibitive to perform grid search, which requires a relatively large number of iterations to find the best-performing hyperparameters. However, to strengthen our argument in line with your suggestion, we are running the grid search in the remaining time and will present the results as soon as they are available.
>
> **References**
>
> [1] Wortsman, et al. Model soups: averaging weights of multiple fine-tuned models improves accuracy without increasing inference time. ICML, 2022.
>
> [2] Weng, R., et al. G-tuning: Improving generalization of pre-trained language models with generative adversarial network. ACL, 2023.
>
> [3] Malladi, S., et al. Fine-tuning language models with just forward passes. NeurIPS 2023.

---

> > ### Comment · Reviewer_JjKU · 2024-08-12
> >
> > Thanks for the response and for running the additional experiments on llama-3 for the empirical observations. I have increased my score from a 6 to a 7.

---

> > > ### Author Response · Authors · 2024-08-12
> > >
> > > Thank you for the positive feedback on our paper. We will incorporate all the discussions into the final manuscript. Also, in your response, you mentioned that you would raise the score to 7, but it seems that it wasn’t updated. Could you please adjust the score accordingly?

---

> > > > ### Comment · Reviewer_JjKU · 2024-08-12
> > > >
> > > > Sorry about that - just updated.

---

### Official Review · Reviewer_WU7w · 2024-07-10

**Soundness:** 3
**Presentation:** 3
**Contribution:** 3
**Rating:** 7
**Confidence:** 4

**Summary:**

The paper presents a novel approach to model fusion through Bayesian Optimization for fine-tuning pre-trained language models on downstream tasks. The authors address the challenges associated with hyperparameter selection and the discrepancy between loss and metric landscapes during the fine-tuning process. They propose a two-stage Bayesian optimization framework that first identifies optimal hyperparameters for fine-tuning and then uses multi-objective Bayesian optimization to find the best combination of models in the parameter space.  The paper provides a comprehensive experimental evaluation to demonstrate the properties and benefits of the proposed approach.

**Strengths:**

1. The paper introduces a novel approach to model fusion through Bayesian Optimization (BOMF), which is a creative application of Bayesian optimization to the problem of fine-tuning pre-trained language models. The originality lies in the development of a two-stage Bayesian optimization process that addresses the specific challenges of hyperparameter tuning and model combination in NLP tasks. The method is innovative as it optimizes both the loss and the desired metrics simultaneously, which is a new perspective in the context of model fusion. The empirical finding that a large discrepancy exists between loss and metric in fine-tuning LLM is valuable to the community.

2. The quality of the paper is evident in its rigorous experimental design and comprehensive evaluation. The authors have tested BOMF across various NLP tasks and models, demonstrating its robustness and effectiveness.

3. The paper is well-structured, with a clear presentation of the problem, the proposed solution, and the experimental setup. Overall, it is well-written and easy to follow.

**Weaknesses:**

1. There are no discussions on computational efficiency: Although the paper mentions a two-stage Bayesian optimization process, it could further elaborate on the computational efficiency of BOMF, especially when compared to traditional fine-tuning methods and the baselines. More detailed complexity analysis or comparisons could help readers understand the practicality of applying BOMF at scale.
2. Hyperparameter sensitivity analysis: The paper could provide a more in-depth analysis of how sensitive BOMF is to the choice of hyperparameters for the Bayesian optimization process itself, e.g., the choice of kernel of Gaussian process regression. Understanding the stability of the method under different settings can be crucial for practitioners.

**Questions:**

See weakness.

**Limitations:**

The authors have adequately discussed the limitations of their work.

---

> ### Author Rebuttal · Authors · 2024-08-07
>
> We appreciate your constructive feedback. We have answered your questions and concerns in this response. Please let us know if you have any follow-up questions.
>
> **[W1] There are no discussions on computational efficiency: More detailed complexity analysis or comparisons could help readers understand the practicality of applying BOMF at scale**
>
> Please refer to the general response for the analysis and additional experiments regarding the computational cost.
>
> **[W2] Hyperparameter sensitivity analysis: The paper could provide a more in-depth analysis of how sensitive BOMF is to the choice of hyperparameters for the Bayesian optimization process itself**
>
> Since the test suites used in this work require huge compute resources, it is difficult to provide hyperparameter sensitive analysis on Bayesian optimization processes. It is noteworthy that the hyperparameters of Bayesian optimization can be thought of as meta-hyperparameters, which should be tuned with multiple tasks and multiple domains. It implies that it necessitates utilizing more compute resources. In addition, from the standpoint of black-box optimization, the sensitive analysis assumes that the specifics of objective functions are known, which is generally infeasible in black-box optimization. Therefore, instead of analyzing the sensitivity of the hyperparameters of Bayesian optimization, we followed their conventional settings. All hyperparameters related to Gaussian processes are optimized via model selection. Matern 5/2 kernel for Gaussian processes and the logarithmic form of expected improvement are used.

---

> > ### Comment · Reviewer_WU7w · 2024-08-12
> >
> > I thank the authors for providing responses to the questions. This rebuttal essentially clarifies my concerns, and I will raise my score to 7 accordingly.

---

> > > ### Author Response · Authors · 2024-08-12
> > >
> > > Thank you for the positive feedback on our paper. We will incorporate all the discussions into the final manuscript.

---

### Official Review · Reviewer_csBg · 2024-07-14

**Soundness:** 3
**Presentation:** 3
**Contribution:** 4
**Rating:** 8
**Confidence:** 4

**Summary:**

This paper proposes "Bayesian Optimization Model Fusion" (BOMF), which is a method to fuse model weights using Bayesian Optimization over a set of metrics for a given target task. The authors motivate the necessity of BOMF for model fusion in large language model fine-tuning by providing evidence that there exists a large misalignment between the loss and metric surface in NLP tasks for pre-trained models. Additionally, the authors demonstrate initially that, for a given target task, the optimal set of hyperparameters for fine-tuning appears to remain consistent across different levels of fine-tuning capacity (# of frozen layers or LoRA rank).

Drawing on these insights, the total proposed method operates in 3 steps:
- First, Bayesian Optimization is used to select the optimal set of hyperparameters for a given task and setting, optimizing the hyperparameters over the target-task metric, rather than validation loss. Additionally, drawing on their earlier observation, the authors perform this BO procedure on a lightweight model (freezing lower layers of the pre-trained model), and then applying the found hyperparameters to the larger model.
- Then, the parameters are fine-tuned on the target-task, and various checkpoints during this fine-tuning procedure are uniformly sampled (towards the end of training) for model fusion.
- Finally, to select the best set of weights to use when fusing the models together (i.e. taking a weighted average of the sampled checkpoint parameters), multi-objective Bayesian Optimization is used to optimize a set of weights that maximizes the Pareto frontier hyper-volume of a set of metrics (as well as the loss) of the target-task validation data.

The authors demonstrate that this procedure results in a fine-tuned model that outperforms standard fine-tuning and other comparable model fusion techniques. Additionally, the authors provide an ablation study, showing the benefit of optimizing for multiple objectives in the model fusion weights, and

**Strengths:**

- While the individual components of the proposed method are not novel in and of themselves, the proposed method results in a new technique for LLM fine-tuning that appears to convincingly improve upon previous relevant methods.
- The authors provide ablation studies for their method, motivating each step of their method.
- The paper provides some novel and interesting insights regarding the alignment between loss and metric surfaces for NLP tasks, and highlights a flaw in loss-based aggregation and fusion methods in LLM fine-tuning, before proposing a solution.

**Weaknesses:**

- The 3 individual components of the proposed method are fairly distinct and disconnected from one another. As a result, it feels as though there are two distinct contributions that are not necessarily related (hyperparameter BO using lower capacity models, and BO for model fusion).
- Because of this disconnect, one of these contributions (hyperparameter transfer using BO) feels less important than the other (Multi-Objective BO for model fusion), resulting in the MOBO for MF feeling under-explored as the "main contribution" (for instance, it would be interesting to see how BOMF works for "model soups", models obtained from different training runs). Instead, both contributions get comparatively equal attention in this work.
- The presentation of the insights used to motivate the proposed method (both the misalignment between metric and loss surface, and the alignment of optimal hyperparameters over capacity) are only showed qualitatively, i.e. over a single dataset and model setting. It is not clear whether these findings generalize well across other NLP tasks, although it could be argued that the success of the proposed method indicates that they generalize well.
- While there are limitations due to the size of the model, unless I am mistaken, all results are taken from a single run, e.g. results are not averaged over different random seeds and it is therefore less clear how meaningful some of the improvements that BOMF makes over it's baselines are.

**Questions:**

n/a

**Limitations:**

Yes

---

> ### Author Rebuttal · Authors · 2024-08-07
>
> We appreciate your constructive feedback. We have answered your questions and concerns in this response. Please let us know if you have any follow-up questions.
>
> **[W1, W2] It feels as though there are two distinct contributions that are not necessarily related. Because of this disconnect, one of these contributions feels less important than the other, resulting in the MOBO for MF feeling under-explored as the “main contribution”.**
>
> To simplify the notations, we will refer to BO-based hyperparameter selection as `outer BO` and BO-based coefficient search as `inner BO`. The bottom line is that ***the outer BO process and the inner BO process are coupled with each other***, so they were not conducted for different purposes.
>
> As outlined in Section 1, BOMF aims to construct the best-performing single model through model fusion within a parameter space. As detailed in Section 4.2, different combinations of fine-tuning hyperparameters (such as learning rate and batch size) yield varied generalization performances after the fine-tuning process. Figure 3 demonstrates that the generalization performances of models before and after fusion align well. Therefore, to identify the best-performing single model after the model fusion, we must first determine the optimal hyperparameters that yield the best-performing single model before the fusion.
>
> For a simpler explanation, let us consider two fine-tuning hyperparameter configurations, H1 and H2. Figure 3 illustrates that if the performance of the fine-tuned solution from the training trajectory with H1 surpasses that of H2, then the performance after the model fusion with fusion members S1, sampled from the trajectory with H1, remains higher than that of the fusion members S2, sampled from the trajectory with H2. This finding, coupled with the non-differentiable nature of metric functions, underscores the necessity of identifying the best hyperparameter configurations through outer BO to achieve BOMF's ultimate objective with inner BO.
>
>
> **[W3] Both the misalignment between metric and loss surface, and the alignment of optimal hyperparameters over capacity are only shown over a single dataset and model setting.**
>
> In our paper, we demonstrated that both full model fine-tuning and LoRA fine-tuning exhibit 1) metric and loss misalignment and 2) optimal hyperparameters alignment even when freeze layers or ranks change. However, we agree that extending these experiments to larger-scale models would strengthen our findings. Therefore, we conducted additional experiments on the Llama-2 model to verify if 1) and 2) still hold.
>
> Firstly, regarding point 2), the alignment of hyperparameters can be easily observed in Figures A.1. For point 1), the misalignment between loss and metrics is evident in Figure A.2. To further demonstrate the misalignment between loss and metrics, as well as among different metrics, we measured Spearman’s correlation using the loss and metrics of 20 sampled models. Also, we empirically demonstrate that BOMF significantly improves this correlation by considering both loss and metrics simultaneously. Please refer to Tables 9, 10, and 11 in Appendix C for full experimental results. We have included a portion of these results here. According to Table R.2 (c), (d), and (e), language tasks involving the Llama model show significantly lower correlation compared to vision tasks (a) and (b). Then, Table R.3 shows that BOMF successfully increases the correlation between loss and metrics compared to other baseline methods.
>
> **[W4] All results are taken from a single run.**
>
> Thank you for your insightful comment regarding the use of random seeds. We agree that running multiple experiments with different seeds is crucial for ensuring the reliability and completeness of the results. Accordingly, we will provide experimental results obtained using multiple seeds for each experiment. Due to computational limitations during the rebuttal period, we were only able to add results for the medium-sized language models in Table 1 in the main paper, using five seeds per method and dataset. This result can be found in Table R.6 and Table R.7.
>
> Additionally, we plan to conduct further experiments on large language models and update all tables with results from multiple seeds in the camera-ready version of this paper.

---

> > ### Comment · Reviewer_csBg · 2024-08-13
> > **Rebuttal Response**
> >
> > Thanks for the response and clarification.
> >
> > > the outer BO process and the inner BO process are coupled with each other
> >
> > I do not mean to suggest that the impact of one does not have an impact on the other, and I agree that the paper clearly demonstrates that if the performance of H1 is greater than H2, then fusion over S1 will be greater than fusion over S2. However, I disagree that they are "coupled" with each other in the sense that we could drop in some other HPO technique in place of the proposed BO procedure and, as long as it results in an effective hyper-parameter search, model fusion would still perform well. In other words, we want to find H2 to do BOMF, but we can find H2 without using BO for hyper-parameter observation.
> >
> > This is more of a stylistic concern however, as the BO for hyper-parameter is still a useful techniques that is backed by empirical observations, and I am not implying that this contribution is not valuable. Only that the paper makes them feel more entangled than they truly are.
> >
> > I think my other listed weaknesses have been addressed, and I appreciate the amount of effort put into the overall rebuttal, which contains results from many new experiments, particularly including results over many random seeds, which I greatly appreciate; I will raise my score to an 8.

---

> > > ### Author Response · Authors · 2024-08-13
> > >
> > > Thank you for the positive feedback on our paper. We will incorporate these discussions into the final manuscript and make the necessary revisions.

---

### Official Review · Reviewer_ut18 · 2024-07-16

**Soundness:** 2
**Presentation:** 3
**Contribution:** 2
**Rating:** 2
**Confidence:** 4

**Summary:**

This paper introduces Bayesian Optimization Model Fusion (BOMF), a method for improving fine-tuning of pre-trained language models. BOMF addresses the challenge of selecting optimal models and hyperparameters by utilizing multi-objective Bayesian optimization to consider both loss and desired metrics during model fusion. It employs a two-stage approach, first optimizing hyperparameters for fine-tuning, then focusing on model fusion. Experiments across various NLP tasks demonstrate significant performance gains using BOMF, showcasing its effectiveness in both Natural Language Understanding and Generation.

**Strengths:**

Using Multi-Objective Bayesian Optimization (MOBO) that considers both metrics and loss functions for model fusion seems like an interesting idea.

**Weaknesses:**

Limited Performance Gain: The observed improvements from the proposed BOMF method are relatively small.

Outdated Models and Datasets: The study relies on older base models (RoBERTa and T5) and simple datasets that may not reflect the current state-of-the-art in terms of model architectures, size, and task complexity.

Incomplete Evaluation for Generative Tasks: The evaluation of generative tasks lacks the use of LLM-based evaluation or human evaluation, which are crucial for assessing the quality and diversity of generated text. Relying solely on computational metrics might not capture the nuances and potential biases of generated outputs.

Questionable Approach: The use of model fusion to address the discrepancy between loss and desired metrics is not necessarily the most effective solution. Alternative approaches like Reinforcement Learning from Human Feedback (RLHF) and Direct Preference Optimization (DPO) can directly optimize towards desired metrics, potentially bypassing the need for fusion altogether.

**Questions:**

1) what if the metrics to be optimized can not be directly computed? How to deal with complex evaluation criteria (e.g. grounding, coherence) commonly seen in LLM evaluation?
2) How the method perform in recent models such as Llama family?

**Limitations:**

Limited Performance Gain: The observed improvements from the proposed BOMF method are relatively small.

Outdated Models and Datasets: The study relies on older base models (RoBERTa and T5) and simple datasets that may not reflect the current state-of-the-art in terms of model architectures, size, and task complexity.

Incomplete Evaluation for Generative Tasks: The evaluation of generative tasks lacks the use of LLM-based evaluation or human evaluation, which are crucial for assessing the quality and diversity of generated text. Relying solely on computational metrics might not capture the nuances and potential biases of generated outputs.

Questionable Approach: The use of model fusion to address the discrepancy between loss and desired metrics is not necessarily the most effective solution. Alternative approaches like Reinforcement Learning from Human Feedback (RLHF) and Direct Preference Optimization (DPO) can directly optimize towards desired metrics, potentially bypassing the need for fusion altogether.

---

> ### Author Rebuttal · Authors · 2024-08-07
>
> We appreciate your constructive feedback. We have answered your questions and concerns in this response. Please let us know if you have any follow-up questions.
>
> **[W1] The observed improvements from the proposed BOMF method are relatively small**
>
> The improvement of BOMF is not negligible compared to other baselines. First, we would like to emphasize that our method consistently outperforms various baselines across a wide range of task types (i.e., classification, question answering, and the medical domain) and models (i.e., T5, LLaMA3). While the degree of improvement might appear marginal in some cases, it is essential to recognize that the extent of improvement, whether small or large, is relative. The fact that our method consistently achieves superior performance in nearly every case serves as an absolute metric that is universally recognizable. Our method consistently outperforms the baselines in these diverse scenarios. Specifically, BOMF achieved a 13%, 14%, 10% error improvement in the GLUE tasks, SQuAD task and KorMedQA tasks, respectively.
>
> Furthermore, the recent fusion research [1,2,3] in various scenarios has shown similar tendency to our results, showing that even if the improvement in specific tasks seems marginal, they exhibit overall superior performance compared to the baseline, proving the necessity of such methods. In particular, even when compared to these methods, our method demonstrates improved performance in almost all experimental results, indicating its effectiveness.
>
> **[W2, Q2] The study relies on older base models (RoBERTa and T5) and simple datasets. How did the method perform in recent models such as the Llama family?**
>
> We have already conducted experiments using Llama-2 and Llama-3 in Sec 6.2 of the main paper, and included the results for summarization, Korean multi-choice medical question answering, and dialogue generation. In most cases of these three experimental settings, our method outperforms other baseline methods. For detailed experimental setting and results, refer to Table 2 in the main text and Tables 15 and 16 in the appendices.
>
>
> **[W3, Q1] The evaluation of generative tasks lacks the use of LLM-based evaluation or human evaluation. How to deal with complex evaluation criteria commonly seen in LLM evaluation?**
>
> Thank you for suggesting additional evaluations on generative tasks. We would like to address this question from two major perspectives:
>
> 1. When a large language model (LLM) is used to evaluate the outputs of various models, BOMF demonstrates superior performance compared to the existing baselines.
>
> 2. BOMF can utilize more complex evaluation criteria such as human evaluation or LLM assessment in the process of model fusion.
>
> Regarding the first one, following your suggestion, we conducted an evaluation with a ChatGPT-turbo-3.5-based approach. This evaluation method involves generating scores by asking the LLM to assess the similarity between the generated responses and the ground-truth answers. Using this, we compared BOMF's performance with other baselines. Here, BOMF refers to the model optimized with the R1, R2, and RL metrics. As shown in Table R.1, even when optimized with this specific metric, BOMF outperformed other baselines. This indicates that our approach not only excels with traditional metrics but also adapts well to the latest evaluation methods.
>
> Additionally, the consideration of both loss and multiple metrics helps the model remain robust across various unseen metrics. Furthermore, since BOMF uses BO to optimize combination coefficients, it requires only the evaluation metric values corresponding to each set of combination coefficients to update them. This means that the optimization process does not rely on a backward process through the metric; as long as evaluation values are available, BOMF can optimize regardless of the complexity of the evaluation procedure. To illustrate this, we conducted optimization using metrics evaluated by LLMs, denoted as the column of ChatGPT BOMF in Table R.1. The result shows that BOMF can optimize the coefficients and achieve performance improvements even with these complex metrics.
>
> **[W4] RLHF and DPO can directly optimize towards desired metrics, potentially bypassing the need for fusion altogether.**
>
> We agree that methods like RLHF can be used to optimize desired metrics. However, we want to emphasize that our method complements rather than contradicts methods like RLHF or DPO. For instance, the hyperparameters used in RLHF or DPO can be optimized more efficiently using BO with a lower rank or fewer layers. Also, when performing RLHF or DPO, multiple checkpoints can be generated during training procedure. Typically, the best validation performance checkpoint is selected from these. However, with BOMF, these checkpoints can be fused to produce better results. By combining multiple checkpoints through BOMF, better generalization performance can be achieved compared to selecting a single checkpoint. This capability is well illustrated in our experiments, as shown in Tables 2 and 4.
> Another important point is that BOMF can consider multiple metrics simultaneously. While methods like RLHF or DPO require detailed preference settings for each metric in different scenarios, BOMF automatically balances various metrics and loss functions to find the optimal combination. To demonstrate that RLHF and BOMF can indeed be combined, we conducted additional experiments. According to Table R.2, performing various fusion methods on RLHF training checkpoints showed that BOMF outperformed other methods.
>
> **References**
>
> [1] Wortsman, et al. Model soups: averaging weights of multiple fine-tuned models improves accuracy without increasing inference time. ICML, 2022.
>
> [2] Weng, R., et al. G-tuning: Improving generalization of pre-trained language models with generative adversarial network. ACL, 2023.
>
> [3] Malladi, S., et al. Fine-tuning language models with just forward passes. NeurIPS 2023.

---

> > ### Comment · Area_Chair_XHQF · 2024-08-13
> >
> > Dear Reviewer,
> >
> > You have not responded to the authors yet.
> > Your review is in very strong disagreement with the other 3 reviews. The points mentioned in both Weaknesses and Limitations are identical and quite generic.
> > To ensure that this is not an LLM-generated review, you need to substantiate your decision and better explain it to the authors.
> >
> > Thank you,
> >
> > Your AC

---

### Author Rebuttal · Authors · 2024-08-07

We thank all reviewers for their valuable and constructive comments.

The reviewers acknowledge that the paper is well-written, easy to follow, and recognize the originality of using Bayesian optimization for model fusion in BOMF (R-ut18, R-csBg, R-WU7w, R-JjKU). Reviewers also recognize the paper for providing a thorough study of this method, motivating each step of the approach, and offering new and interesting insights into the alignment between loss and metric surfaces for NLP tasks (R-csBg, R-WU7w, R-JjKU). Additionally, reviewers agree that the authors clearly demonstrate the quality of the paper through rigorous experimental design and comprehensive evaluation. They acknowledge that the authors test BOMF across various NLP tasks and models, proving its robustness and effectiveness (R-csBg, R-WU7w).
All supplementary experimental outcomes and discussions will be incorporated into the final manuscript.

**Regarding the computational efficiency of BOMF**

In response to the reviewers' concerns regarding the computational efficiency of BOMF, we analyze these points as follows:

The additional computational cost incurred by BOMF can be analyzed in two ways: 1) in comparison to basic fine-tuning and 2) relative to other fusion baseline methods.

First, compared to basic fine-tuning, the computational cost during hyperparameter tuning in BOMF is similar to that of grid search in fine-tuning. Given that we used the same number of iterations for the BO step and the same number of grid points for the grid search method. While grid search can theoretically reduce search time through parallelization with sufficient GPUs, the increasing memory and computational demands of large language models make it challenging to perform parallel grid searches without a significant number of GPUs. Consequently, the time difference between BOMF and basic fine-tuning may not be substantial. Thus, the additional computational cost of BOMF mainly consists of the cost associated with calculating the combination coefficient. Specifically, during the MOBO step, if performed K times, the additional computational cost includes K evaluations of the validation set metrics and losses, as well as the computations required for Bayesian Optimization.

When compared to other fusion methods, BOMF, like other baselines (excluding SWA), requires multiple forward passes to find the best combination coefficient. Methods like learned SWA, which optimize combination coefficients via backward passes, incur additional backward computation costs.

*Table R.5* shows the empirical results comparing the time required for combination coefficient optimization between BOMF and other fusion baselines. The experiments were conducted using the Llama-3 model on a Korean multi-choice medical question answering task to demonstrate the scenario with a large language model.

Our additional experiments are included in the below PDF.

---

### Decision · Program_Chairs · 2024-09-25

**Decision:**

Accept (spotlight)

**Comment:**

The paper presents a novel approach to model fusion through Bayesian Optimization for fine-tuning pre-trained language models on downstream tasks. The authors address the challenges associated with hyperparameter selection and the discrepancy between loss and metric landscapes during the fine-tuning process. They propose a two-stage Bayesian optimization framework that first identifies optimal hyperparameters for fine-tuning and then uses multi-objective Bayesian optimization to find the best combination of models in the parameter space. The paper provides a comprehensive experimental evaluation to demonstrate the properties and benefits of the proposed approach.

This paper has 3 accept scores (7, 7, 8) and 1 strong reject (2).

Let me start with the strong reject: the reviewer (ut18) has not responded to the authors rebuttal and my repeated requests (3) for feedback. His review is so generic and disconnected from the actual work that I suspect it is LLM-generated. Among the  “weaknesses” mentioned in the review, this one is especially meaningless:
“Questionable Approach: The use of model fusion to address the discrepancy between loss and desired metrics is not necessarily the most effective solution. Alternative approaches like Reinforcement Learning from Human Feedback (RLHF) and Direct Preference Optimization (DPO) can directly optimize towards desired metrics, potentially bypassing the need for fusion altogether.”
This work is about extracting the best model from a fine-tuning training run and is orthogonal to the source of supervision. Applying it to the supervised tuning phase of medium or large language models is the obvious priority. To please reviewer ut18, the authors also demonstrated their approach on RLHF, but there was no response.

Thus, this meta-review will only focus on the 3 accept reviews, all of them of high quality, with responses to the author’s rebuttal.
Among the strength, the reviewers mention that the paper is well-written and mention the originality of the approach, even if the individual components of the proposed method are not novel. The paper is  thorough in its methodology and experiments, motivates each step of the approach, and offers novel and interesting insights into the alignment between loss and metric surfaces for NLP tasks.

Among the weaknesses, there were concerns about the computational efficiency of the method and the small size of the models. They were addressed in the rebuttal, with new results: performance on LLAMA-3, time cost, application to RLHF, etc…
Remarkably, all 3 reviewers increased their score after this rebuttal.

This well written paper proposes an original method with significant improvements over basic fine-tuning, or other model averaging methods. The computational cost is reasonable, so the impact could be significant